# Functional interactions among neurons within single columns of macaque V1

**Ethan B Trepka[1,2†], Shude Zhu[1]\*†, Ruobing Xia[1], Xiaomo Chen[1,3], Tirin Moore[1]**

[1]Department of Neurobiology, Howard Hughes Medical Institute, Stanford University, Stanford, United States; [2]Neurosciences Program, Stanford University, Stanford, United States; [3]Center for Neuroscience, Department of Neurobiology, Physiology, and Behavior, University of California, Davis, Davis, United States

**\*For correspondence:**
shude@stanford.edu

†These authors contributed equally to this work

**Abstract** Recent developments in high-density neurophysiological tools now make it possible to record from hundreds of single neurons within local, highly interconnected neural networks. Among the many advantages of such recordings is that they dramatically increase the quantity of identifiable, functional interactions between neurons thereby providing an unprecedented view of local circuits. Using high-density, Neuropixels recordings from single neocortical columns of primary visual cortex in nonhuman primates, we identified 1000s of functionally interacting neuronal pairs using established crosscorrelation approaches. Our results reveal clear and systematic variations in the synchrony and strength of functional interactions within single cortical columns. Despite neurons residing within the same column, both measures of interactions depended heavily on the vertical distance separating neuronal pairs, as well as on the similarity of stimulus tuning. In addition, we leveraged the statistical power afforded by the large numbers of functionally interacting pairs to categorize interactions between neurons based on their crosscorrelation functions. These analyses identified distinct, putative classes of functional interactions within the full population. These classes of functional interactions were corroborated by their unique distributions across defined laminar compartments and were consistent with known properties of V1 cortical circuitry, such as the lead-lag relationship between simple and complex cells. Our results provide a clear proof-of-principle for the use of high-density neurophysiological recordings to assess circuit-level interactions within local neuronal networks.

## Editor's evaluation

This is an important paper which shows how high-density neurophysiological recordings in non-human-primates can be used to identify inter-neuronal interactions based on cross-correlations. This provides valuable insights such as the dependence of correlations on vertical distance and orientation tuning. Overall the techniques used here are compelling and set a standard for recordings in non-human-primates. The paper is of interest for a broad audience of neuroscientists that performs electrophysiological recordings or is interested in functional interactions among neuron pairs.

## Introduction

Understanding the functional logic of local neuronal microcircuits is among the more fundamental objectives in the study of neural systems, yet it is also among the most challenging. This seems particularly true for mammalian neocortical circuits involved in perceptual and cognitive functions, and most notably in nonhuman primate model systems for which the available tools to interrogate those circuits are the most limited. The columnar organization of the mammalian neocortex (*Lorente de No, 1938*; *Mountcastle, 1957*) and its distinctly layered structure within different cortical domains

are both widely appreciated (Reviewed in *DeFelipe et al., 2012*; *Horton and Adams, 2005*; *Mountcastle, 1997*). In addition, several key principles of cortical circuitry, including constituent cell types (*Harris and Mrsic-Flogel, 2013*; *Jiang et al., 2015*; *Kätzel et al., 2011*; *Markram et al., 2004*; *Network, 2021*; *Packer and Yuste, 2011*; *Yoshimura and Callaway, 2005*), input-output organization (*Callaway, 1998*; *Douglas and Martin, 2004*; *Lefort et al., 2009*; *Muñoz-Castañeda et al., 2021*; *Thomson and Bannister, 2003*; *Weiler et al., 2008*) and local microcircuit motifs (*Avermann et al., 2012*; *Frandolig et al., 2019*; *Karnani et al., 2016*; *Obermayer et al., 2018*; *Pfeffer et al., 2013*; *Pi et al., 2013*) have emerged in recent years. Although it remains to be determined, such principles may turn out to generalize not only across neocortical areas, but also across species (*Harris and Shepherd, 2015*; *Karten, 2015*; *Stacho et al., 2020*) (see also *Campagnola et al., 2022*; *Wildenberg et al., 2021*). Yet, mapping complete cortical microcircuits within even a single cortical area remains a tremendous challenge (*Adesnik and Naka, 2018*).

Recent advances in recording technology have facilitated the development of large-scale, high-density micro-electrode arrays resulting in a substantial increment (>10x) in the number of neurons that can be studied simultaneously within a localized area of neural tissue. A prime example is the recent development of the Neuropixels probe (IMEC, Inc), which consists of a high-channel count Si shank with continuous, dense, programmable recording sites (~1000/cm). Numerous recent studies have demonstrated the advantages of such probes, such as their use in recording large neuronal populations within deep structures where optical approaches cannot be deployed (*Jun et al., 2017*; *Steinmetz et al., 2019*). In addition, the high-density capacity of such probes dramatically increases the quantity of single neurons that can be obtained within a localized area of neural tissue (*Siegle et al., 2021*), thus making them well-suited for investigations of local neuronal circuitry. Given that studies of local neuronal circuitry within the primate brain are notoriously difficult to achieve, high-density electrophysiological approaches may be particularly valuable. However, only a few electrophysiological studies of the primate brain using such probes have been carried out thus far (*Hesse and Tsao, 2020*; *Paulk et al., 2022*; *Sun et al., 2022*; *Trautmann et al., 2019*; *Zhu et al., 2020*).

To date, many studies have leveraged the covariation in spiking activity between simultaneously recorded neurons to elucidate underlying neural mechanisms in the primate brain with some success, particularly within the visual system (*Briggs et al., 2013*; *Chu et al., 2014*; *Hansen et al., 2012*; *Hembrook-Short et al., 2019*; *Jia et al., 2013*; *Kohn and Smith, 2005*; *Koren et al., 2020*; *Krüger and Aiple, 1988*; *Maldonado et al., 2000*; *Smith and Kohn, 2008*; *Zandvakili and Kohn, 2015*). In particular, temporally precise correlations in spiking activity have provided a unique means of assessing interactions among neurons in both local and distributed networks (*Aertsen and Gerstein, 1985*; *Aertsen et al., 1989*; *Diba et al., 2014*; *Moore et al., 1970*; *Nelson et al., 1992*; *Nowak et al., 1999*; *Perkel et al., 1967*; *Siegle et al., 2021*), and identification of such interactions has played an important part in understanding neural circuits in the mammalian visual system (*Alonso and Martinez, 1998*; *Alonso et al., 1996*; *Alonso et al., 2001*; *Baker and Bair, 2012*; *Cohen and Kohn, 2011*; *Das and Gilbert, 1999*; *Denman and Contreras, 2014*; *Michalski et al., 1983*; *Nelson et al., 1992*; *Reid and Alonso, 1995*; *Schwarz and Bolz, 1991*; *Senzai et al., 2019*; *Siegle et al., 2021*; *Toyama et al., 1981a*; *Ts'o et al., 1986*; *Usrey et al., 1998*; *Usrey et al., 1999*). However, the extent of circuit-level details addressable with crosscorrelation is greatly limited by the low incidences of simultaneous recordings from connected neurons when using conventional extracellular recording techniques (e.g. *Alonso et al., 2001*; *Hembrook-Short et al., 2019*; *Nelson et al., 1992*; *Ts'o et al., 1986*). The use of high channel-count probes should substantially mitigate that limitation by virtue of the large increment in recording yield. Moreover, the high-density of recordings should further increase the incidence of identifiable correlated neuronal activity by virtue of the proximity of recorded cells.

We assessed the capacity of high-density Neuropixels probes to identify functional interactions among pairs of neurons within cortical columns of primary visual cortex of macaque monkeys using established crosscorrelation approaches. Crosscorrelation assesses the statistical dependencies between spike trains of two or more neurons and has long played an important role in estimating how ensembles of neurons interact with one another (*Casile et al., 2021*; *Okatan et al., 2005*; *Perkel et al., 1967*). Although a number of different analyses have been employed (*Casile et al., 2021*; *Keeley et al., 2020*; *Kobayashi et al., 2019*), crosscorrelation is the most widely used, perhaps due to its simplicity. Significant crosscorrelations are broadly interpreted as identifying 'functional

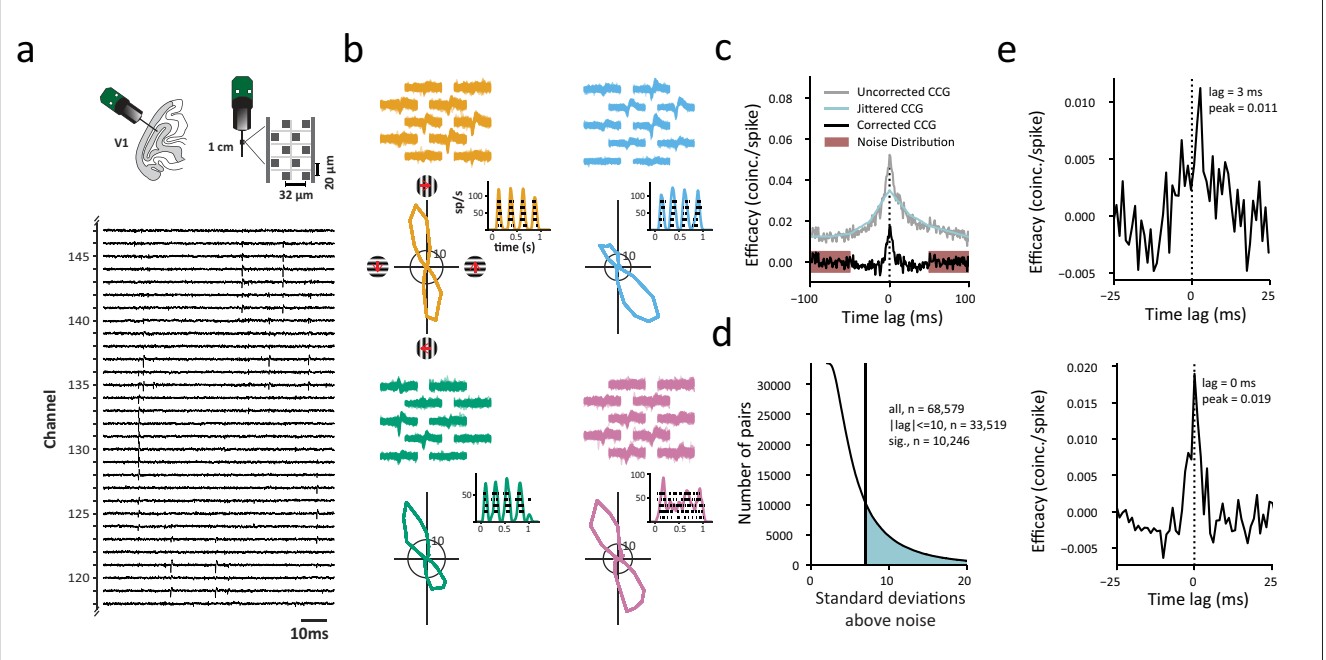

**Figure 1.** Identifying functional interactions within single columns of visual cortex. (**a**) Upper left cartoon depicts the angle of Neuropixels probe penetrations made into the lateral surface and underlying calcarine sulcus of V1. Upper right, Neuropixels probe base and shank, and layout of electrode contacts for a section of the recording shank. Lower, raw voltage traces recorded from an exemplar section of channels and time period. (**b**) Example single-neuron recordings with Neuropixels probes, three simple cells (orange, blue, green) and one complex cell (purple). Top, spike waveforms recorded across multiple adjacent electrode contacts are shown for each neuron. Bottom, each neuron's response to its preferred orientation (rasters and instantaneous spike rates) and their corresponding tuning curves. Red arrows (upper left) denote the drift direction of oriented gratings. (**c**) Example CCG between an example pair of V1 neurons. Corrected CCGs were generated from the difference between a jittered and an uncorrected CCG. Significance of each CCG was determined from comparisons between the peak and the noise distribution. (**d**) Distribution of ratios of CCG peaks to the noise (SD) for all recorded pairs. Shaded area denotes CCGs with peaks >7 SDs above the mean of the noise distribution. (**e**) Two example CCGs differing in both peak lag and peak efficacy.

connections' (**Aertsen and Gerstein, 1985**; **Dann et al., 2016**; **Denman and Contreras, 2014**; **Menz and Freeman, 2004**; **Schwarz and Bolz, 1991**) or 'functional interactions' among neurons (**Gochin et al., 1991**; **Ostojic et al., 2009**), presumably to distinguish them from more direct measurements of synaptic effects. These functional interactions are interpreted as reflecting one of myriad putative circuit arrangements among neuronal pairs (**Aertsen and Gerstein, 1985**; **Melssen and Epping, 1987**; **Moore et al., 1970**; **Ostojic et al., 2009**), arrangements which include direct monosynaptic connections (**Alonso and Martinez, 1998**; **English et al., 2017**; **Reid and Alonso, 1995**) or neurons with common input (**Constantinidis et al., 2001**; **Das and Gilbert, 1999**; **Denman and Contreras, 2014**; **Michalski et al., 1983**; **Ts'o et al., 1986**; **Türker and Powers, 2001**), either of which can provide important insight into local network architecture. In this study, we identified 1000s of functionally interacting neuronal pairs during single recordings from neurons situated in different cortical layers. Our results demonstrate robust, systematic variations in the synchrony and strength of functional interactions within cortical columns. In addition, by leveraging the large numbers of interacting pairs, distinct classes of interactions could be identified within the full population.

## Results
### Identifying functional interactions within single columns of visual cortex
The activity of V1 neurons was recorded in two anesthetized macaque monkeys (M1, M2) using high-density, multi-contact Neuropixels probes (version 3A; IMEC Inc, Belgium; *Figure 1a*; Methods). Each probe consisted of 986 contacts (12 mm x 12 mm, 20 µm spacing) distributed across 10 mm, of which 384 contacts could be simultaneously selected for recording. Probes were inserted into the lateral operculum of V1 with the aid of a surgical microscope at angles nearly perpendicular to the cortical

surface. The dense spacing between electrode contacts provided multiple measurements of the waveforms from individual neurons (mean = 4.52 measurements) (*Figure 1a and b*) and facilitated the isolation of large numbers of single neurons. In each of the 5 experimental sessions (3 in M1, 2 in M2), we measured the visual responses of 115–221 simultaneously recorded neurons to drifting gratings presented at varying orientations (total = 802 neurons). As expected, neurons were highly orientation selective, and exhibited both simple and complex cell properties (*De Valois et al., 1982*; *Hubel and Wiesel, 1962*; *Hubel and Wiesel, 1968*; *Figure 1b*). The ratio of simple to complex neurons, respectively, was 1:2.4; 236/802 neurons were simple, and 566/802 neurons were complex. As in previous studies (*Briggs et al., 2013*; *Hembrook-Short et al., 2019*; *Jia et al., 2013*; *Kohn and Smith, 2005*; *Siegle et al., 2021*; *Smith and Kohn, 2008*; *Zandvakili and Kohn, 2015*), we used the visually driven spike trains to measure crosscorrelations between simultaneously recorded neuronal pairs.

To estimate the functional interactions between pairs of neurons recorded simultaneously within columns of V1, we computed cross-correlograms (CCGs) using the 802 visually responsive neurons recorded across sessions. CCGs were computed from the spike trains of 68,579 pairs of simultaneously recorded neurons (6,555–24,310 pairs/session, Methods). Each CCG was normalized by the firing rate (FR) and jitter-corrected to mitigate the influences of FR (*Bair et al., 2001*; *Mastronarde, 1983*) and correlated slow fluctuations (*Harrison and Geman, 2009*; *Smith and Kohn, 2008*), respectively, yielding a corrected CCG (*Figure 1c*). In addition, as in previous studies, we considered a CCG significant only if its peak occurred within 10ms of zero time lag, and if that peak was >7 standard deviations above the mean of the noise distribution (*Siegle et al., 2021*). Using these criteria, a total of 10,246 significant CCGs were obtained from all recording sessions (*Figure 1d*), with each session yielding 755–3,022 significant CCGs. The peak lag of each CCG, defined as the differences between zero and the time when the peak occurred, estimates the synchrony and/or direction of functional interactions between neuronal pairs; whereas the peak efficacy measures the strength of interactions (*Figure 1e*).

## Variation in the synchrony and strength of functional interactions within cortical columns

A number of previous studies using low-channel count probes or chronically implanted electrode arrays have shown that correlated activity in primate V1 declines with the horizontal distance separating pairs of neurons (*Krüger and Aiple, 1988*; *Maldonado et al., 2000*; *Smith and Kohn, 2008*) (see also *Chu et al., 2014*). Evidence from these studies suggest that correlations are greatest for pairs of neurons located within the same column, and diminish with greater columnar distance. Other evidence shows variation in the spike timing correlations between neuronal pairs located within different laminar compartments (*Smith et al., 2013*). However, considerably less is known about how the nature of correlations varies across the depth of individual columns where the degree of shared input and connectivity is at its highest. We therefore leveraged the large numbers of significantly correlated pairs obtained from high-density recordings to examine how the synchrony and strength of correlations depended on the vertical distance separating neurons within V1 columns. *Figure 2a* shows data from an example recording session in which 221 visually responsive neurons were recorded and 2,453 significantly correlated pairs were obtained. All neurons are shown along the ~2 mm depth of cortex. Shown also are two example correlated pairs whose CCGs are shown in *Figure 1e*. Of the two pairs, the vertical distance separating neurons in one pair was 138 µm greater than that of the other. In spite of this small difference, the CCG of the closer pair was both more synchronous and stronger than the more distant pair. This pattern of results was observed across all significantly correlated pairs and in all sessions (*Figure 2b–c*; *Figure 2—figure supplements 1–2*). The synchrony of correlated spiking diminished several fold across neuronal pair distance. This change could be fit with a linear function ($r$=0.42; $p<10^{-5}$) in which the (absolute) peak lag increased at a rate of 1.3 ms / 500 µm of vertical distance. Peak efficacy of the significant CCGs also depended heavily on pair distance. This effect could be fit with an exponential decay function ($r$=–0.34; $p<10^{-5}$) in which the peak efficacy decreased by half within 154 µm. Thus, both measures of functional interactions depended heavily on the vertical distance separating neuronal pairs. In addition, we confirmed that the effects of vertical distance on both the synchrony and strength of CCGs were independent of whether neuronal pairs were located within the same or different cortical layers (*Figure 2—figure supplement 3*).

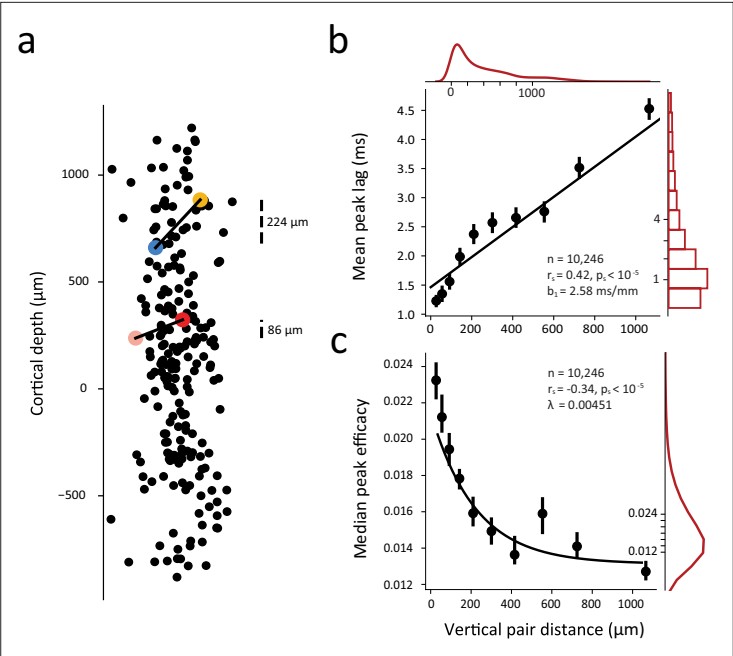

**Figure 2.** Dependence of synchrony and strength of functional interactions on vertical distance within single cortical columns. (**a**) Example session (M1, session 3) with 221 visually responsive neurons recorded simultaneously and their locations across cortical depth. (Horizontal axis is magnified for visualization). Cortical depth 0 denotes the boundary between Layer 4c and Layer 5. Laminar boundaries were determined using histological data and current-source-density (CSD) profile for each session (Methods). Two example correlated pairs from **Figure 1e** with varied CCGs are shown in color (blue-yellow pair and pink-red pair corresponding to **Figure 1e** top and bottom, respectively). (**b**) Linear dependence of synchrony on vertical pair distance. (**c**) Strength of CCGs decay with greater pair distance. In b and c, all significantly correlated pairs from all sessions are combined and each dot denotes the mean peak lag or median peak efficacy of significantly correlated CCGs within a (10% quantile) vertical distance bin. Error bars denote 95% confidence intervals. Black lines denote the linear and exponential fits in b and c, respectively; slope (b) and decay constant ($\lambda$) are shown. Red lines and bar plots show marginal distributions.

The online version of this article includes the following figure supplement(s) for figure 2:

**Figure supplement 1.** Dependence of synchrony on vertical distance within single cortical columns for all individual sessions.

**Figure supplement 2.** Dependence of the strength of functional interactions on vertical distance within single cortical columns for all individual sessions.

**Figure supplement 3.** Dependence of the synchrony and strength of functional interactions on vertical distance within single cortical columns, separated by laminar pairings.

## Dependence of synchrony and strength of functional interactions on tuning similarity

In addition to the dependence of correlated activity on the distance between neuronal pairs, many studies have shown that greater functional and synaptic connectivity typically occurs between neurons with similar stimulus preferences (*Chu et al., 2014*; *Constantinidis et al., 2001*; *Cossell et al., 2015*; *DeAngelis et al., 1999*; *Denman and Contreras, 2014*; *Funahashi and Inoue, 2000*; *Lee et al., 2016*; *Ts'o et al., 1986*) (but see *Das and Gilbert, 1999*; *Maldonado et al., 2000*). Within primate V1, stimulus selectivity is notably similar for neurons within the same column, particularly for orientation selectivity (*Blasdel and Salama, 1986*; *Hubel and Wiesel, 1968*; *Hubel and Wiesel, 1974*; *Ts'o et al., 1990*), and this was evident in our recording sessions, where the peak visual responses were largely aligned at the same stimulus orientation across cortical depth (*Figure 3a*). We considered that within orientation columns, functional interactions could be homogenous for populations of similarly tuned neurons. Alternatively, it could be that even small variations in tuning similarity could result in robust differences in the synchrony and strength of correlated activity. To address this, we examined the dependence of synchrony and strength on the similarity of visual properties of neurons within the same

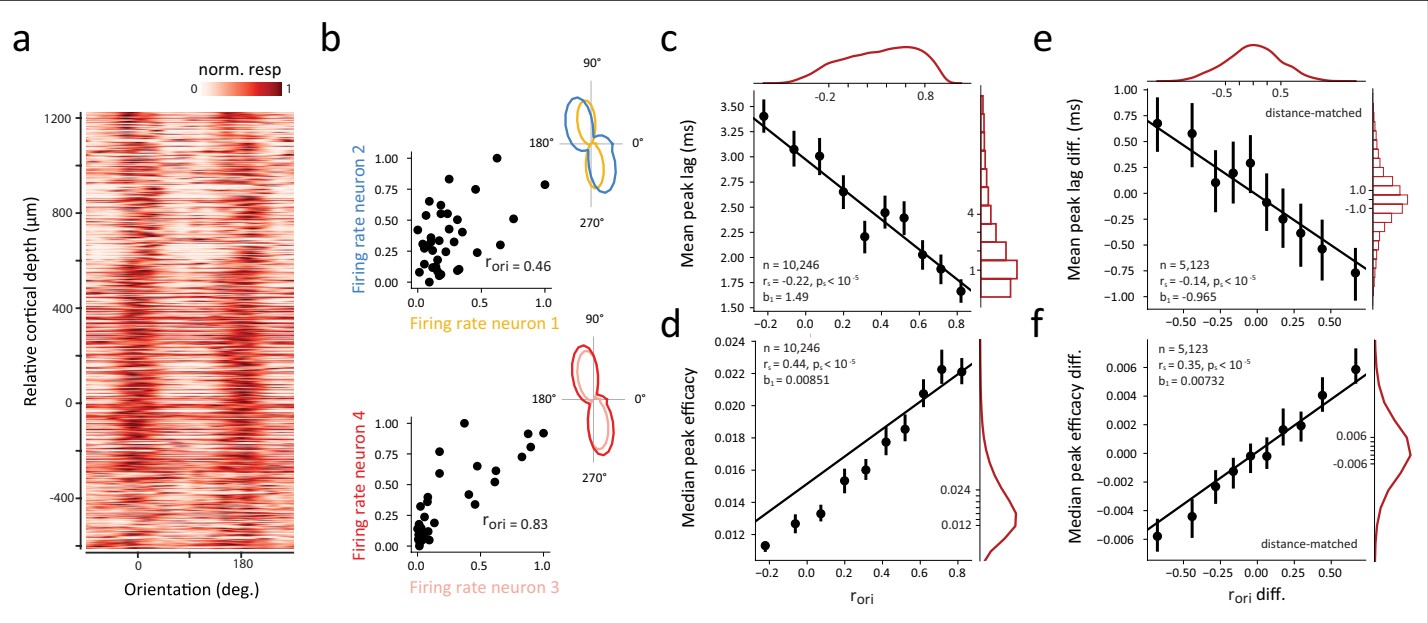

**Figure 3.** Dependence of synchrony and strength of functional interactions on tuning similarity within single cortical columns. (**a**) Heat map of visual responses across drift directions of oriented gratings and across cortical depth. The response tuning of each of 802 neurons was aligned to the overall preferred orientation shared by neurons recorded from the same session, and all sessions were combined. (**b**) Signal correlation between exemplar neurons. Left, Scatter plot of normalized responses to different stimulus orientations (n=36) for the two example pairs shown in *Figure 1e* and *Figure 2a*. Signal correlations ($r_{ori}$) are also shown. Right, each neuron's orientation tuning curve. (**c**) Linear dependence of synchrony on the corresponding signal correlation. (**d**) Linear dependence of CCG strength on the corresponding signal correlation. (**e**) Difference in peak lag of distance-matched CCGs was negatively correlated with difference in signal correlation. (**f**) Difference in peak efficacy of distance-matched CCGs was positively correlated with difference in signal correlation. In c-f, all significantly correlated pairs from all sessions are combined and each dot denotes mean peak lag or median peak efficacy of significantly correlated CCGs within a (10% quantile) signal correlation bin. Error bars denote 95% confidence intervals. Black lines denote linear fits; slopes (b) are shown. Red lines and bar plots show marginal distributions.

The online version of this article includes the following figure supplement(s) for figure 3:

**Figure supplement 1.** Dependence of synchrony on tuning similarity within single cortical columns for all individual sessions.

**Figure supplement 2.** Dependence of the strength of functional interactions on tuning similarity within single cortical columns for all individual sessions.

cortical column. As in previous studies (*Shadlen and Newsome, 1998*; *Zohary et al., 1994*), we quantified tuning similarity by computing signal correlations ($r_{ori}$) for each neuronal pair (Methods). Across the total number of neuronal pairs (N=68,579), the mean $r_{ori}$ was 0.25. For the significantly correlated neuronal pairs, the mean $r_{ori}$ was 0.33. Signal correlations for the two previous example neuronal pairs are shown in *Figure 3b*. The responses of both pairs are positively correlated, yet that correlation is much higher in the second, more proximal, pair (*Figure 2a*) and the one with a more synchronous and stronger CCG (*Figure 1e*). Overall, we found that both the peak lag and peak efficacy of CCGs for significantly correlated neuronal pairs varied monotonically with tuning similarity across the range of signal correlations (*Figure 3c and d*). Neuronal pairs with the highest signal correlations exhibited half the peak lags and twice the peak efficacies of uncorrelated pairs. This pattern was observed in each of the individual recording sessions (*Figure 3—figure supplements 1–2*).

We considered that the apparent relationship between the synchrony and strength of functional interactions and signal correlation might result indirectly from a collinear effect of vertical distance on CCGs (*Figure 2*). To address this, we examined differences in the peak lags and peak efficacies of CCGs between combinations of two neuronal pairs separated by comparable cortical distances. Specifically, we sorted all significantly correlated CCGs by their vertical distances, and then examined whether differences in signal correlations ($r_{ori}$) among adjacently sorted (distance-matched) pairs were still associated with differences in CCG peak lags and peak efficacies (Methods). Indeed, we found that the differences in peak lags of distance-matched CCGs were negatively correlated with signal correlation (*Figure 3e*) and the differences in peak efficacies of distance-matched CCGs were

positively correlated with signal correlation (*Figure 3f*). These results indicate that signal correlations within the column predicted both the synchrony and strength of functional interactions independent of vertical pair distance. Nonetheless, the distance-matched correlations (*Figure 3e–f*) were smaller than their corresponding unmatched correlations (*Figure 3c–d*), suggesting that the vertical distance between neurons and their orientation signal correlations exhibit distinct, but overlapping, effects on the timing and strength of functional interactions within a single cortical column.

To quantify the distinct contributions of vertical pair distance and orientation signal correlation to the synchrony and strength of CCGs, we fit GLMs to predict CCG peak lag and peak efficacy using pair distance and signal correlation as predictors (Methods). Predictors were standardized (z-scored) so that their relative effects could be compared, and peak outliers (1.5*IQR criterion) were removed. The resulting regression equations were:

$$peak\,lag = 2.48 + 0.94 \times pair\,dist. - 0.37 \times r_{ori}$$
$$peak\,efficacy = 0.018 - 0.0017 \times pair\,dist. + 0.0029 \times r_{ori}$$

Regressions explained 19% of variance in peak lag ($R^2 = 0.191$) and 20% of variance in peak efficacy ($R^2 = 0.195$). Because predictors were standardized, the regression coefficients capture the change in peak lag/efficacy associated with a 1 standard deviation (SD) increase in pair distance/signal correlation. In the regression predicting CCG peak lag, a 1 SD increase in pair distance was associated with a 0.94ms increase in peak lag whereas a 1 SD increase in signal correlation was associated with a 0.37ms decrease in peak lag. Thus, for CCG peak lag, the coefficient of pair distance was nearly three times the coefficient of signal correlation. In contrast, for CCG peak efficacy, the coefficient of signal correlation was nearly twice that of pair distance. Thus, whereas signal correlation was less predictive of CCG peak lag, it was more predictive of CCG peak efficacy than pair distance.

## Classification of functional interactions

CCG peak lags and peak efficacies are often the parameters of interest in cross correlations (*Briggs et al., 2013*; *Hembrook-Short et al., 2019*; *Smith and Kohn, 2008*), yet they are simplifications of the more complex, underlying crosscorrelation functions. The shape of these correlation functions may offer additional insights into the distinct types and properties of functional interactions present among neurons within a network. Several theoretical studies have suggested a correspondence between CCG shape and underlying pairwise connectivity (*Aertsen and Gerstein, 1985*; *Melssen*

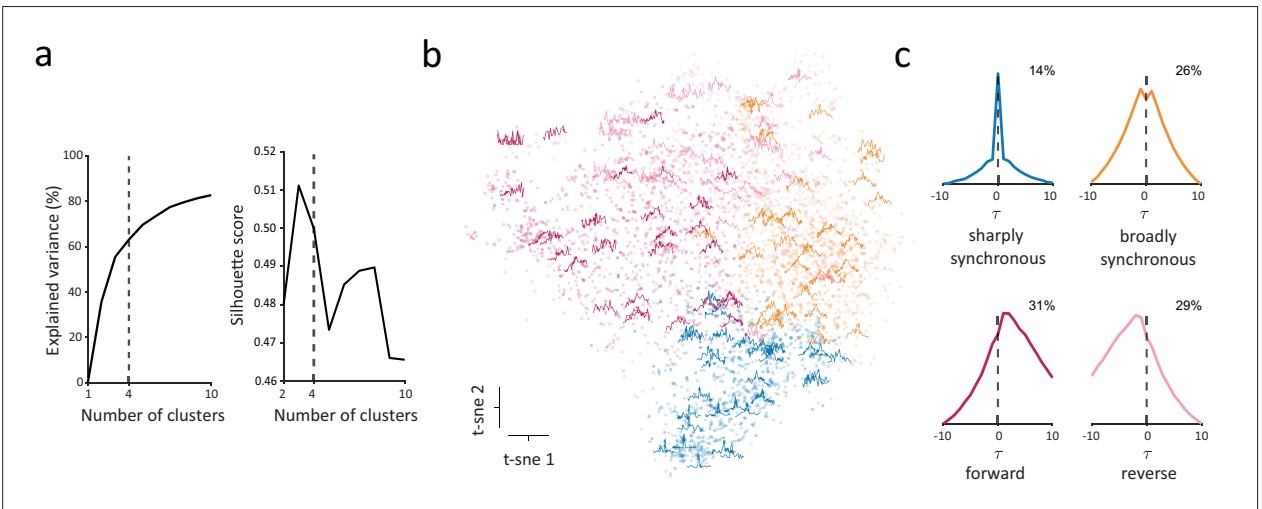

**Figure 4.** Identification of distinct classes of functional interactions within the full population. (**a**) Explained variance (left) and silhouette score (right) as a function of number of clusters. The dashed vertical line indicates the selected number of clusters (n=4). (**b**) Scatter plot of dimensionality-reduced CCGs in the first two dimensions of t-SNE space. Randomly selected example CCGs are overlayed on the scatterplot in their corresponding location in t-SNE space. (**c**) CCG templates generated by averaging over all the CCGs in each cluster. The templates include a 'sharply synchronous' class ($S_{sync}$) with a narrow peak at $\tau = 0$, a 'broadly synchronous' class ($B_{sync}$) with a wide peak at $\tau = 0$, a 'forward' class ($F_{async}$) (leading) with more probability density after $\tau = 0$, and a 'reverse' class ($R_{async}$) (lagging) with more probability density before $\tau = 0$. Numbers denote the percentage of each class among all significantly correlated pairs.

*and Epping, 1987*) that can be further influenced by overall network structure and background noise (*Ostojic et al., 2009*). For example, synchronous CCGs tend to correspond to pairs of neurons that receive input from a common source, while asynchronous CCGs tend to correspond to pairs that have direct synaptic connections (*Ostojic et al., 2009*). Moreover, synchronous CCGs with narrow peaks and synchronous CCGs with broad peaks may correspond to pairs of neurons that receive input from common sources with shorter and longer autocorrelation timescales, respectively (*Ostojic et al., 2009*). Experimental studies have corroborated these findings and identified similar CCG shapes in different cortical regions and species (*Alonso and Martinez, 1998*; *Constantinidis et al., 2001*; *Hembrook-Short et al., 2019*; *Siegle et al., 2021*). However, the distribution of these CCG shapes within a single cortical column remains unknown. Furthermore, whether that distribution within V1 corroborates other evidence about the functional and/or anatomical relationships among V1 laminae and cell types remains unclear.

To address these questions, we clustered the entire population of CCGs, taking advantage of the large number of significantly correlated pairs to identify robust CCG templates. To do this, we first normalized significant CCGs, and utilized t-distributed Stochastic Neighbor Embedding (t-SNE) to map CCGs to a lower dimensional space, and then clustered CCGs in the resulting space using k-means (Methods). To select a statistically reasonable number of clusters, we examined how the total variance explained by clustering and the silhouette score changed as a function of the number of clusters (*Figure 4a*). From this, we selected four as the optimal number of clusters given that silhouette score peaked ~3–4 clusters, and 4 clusters explained more variance than 3.

CCG shape was relatively heterogenous within each of the four clusters (*Figure 4b*). Nonetheless, by averaging over all CCGs in each cluster, we could construct CCG templates that summarized key characteristics of the clusters (*Figure 4c*). Within the full population, we identified two synchronous classes of functional interactions, a 'sharply synchronous' class ($S_{sync}$) with a narrow peak at $\tau = 0$ and a 'broadly synchronous' class ($B_{sync}$) with a wide peak at $\tau = 0$. In addition, two asynchronous classes were identified, a 'reverse' class ($R_{async}$) (lagging) and a 'forward' class ($F_{async}$) (leading) with more probability density before and after $\tau = 0$ (median $\tau = 3ms$), respectively (*Figure 4c*). Aside from clear differences in peak lags between subsets of the putative classes (e.g. synchronous vs. asynchronous), CCGs of different classes also differed in their peak efficacies; synchronous classes exhibited higher average peak efficacies than asynchronous classes (median peak efficacy: $S_{sync}$ 0.021, $B_{sync}$ 0.020, $F_{async}$ 0.015, $R_{async}$ 0.014). Importantly, our objective was not to find the exact number of distinct classes of functional interactions in V1 or to perfectly categorize every interaction into a homogenous cluster. Instead, we sought to identify at least one set of clusters that is consistent with that expected in local microcircuits.

## Corroboration of putative CCG classes with V1 microcircuitry

We next examined the extent to which the putative CCG classes were also distinguishable from one another along anatomical and functional lines given other known properties of V1 microcircuits. First, we considered that the identified classes might differ in their vertical pair distances and signal correlations. Indeed, we found that vertical pair distances were larger and orientation signal correlations were smaller in asynchronous ($F_{async}$ and $R_{async}$) than in synchronous ($S_{sync}$ and $B_{sync}$) classes (*Figure 5a–b*) (significant pairwise comparisons: $p<10^{-5}$). Given that both the peak lag and peak efficacy components of CCGs were clearly predicted by distance and signal correlation (*Figures 2–3*), the observed difference between the synchronous and asynchronous classes is expected. However, additional differences emerged between the identified synchronous classes. For example, we found that in spite of exhibiting similar CCG peak lags, $B_{sync}$ pairs were separated by greater vertical distances than $S_{sync}$ ones (*Figure 5a*; $p<10^{-5}$). Furthermore, in spite of being separated by a greater cortical distance, $B_{sync}$ pairs exhibited higher signal correlations than $S_{sync}$ pairs (*Figure 5b*; $p<10^{-5}$). These findings thus provide some validation of the apparent subtypes of CCGs.

Next, we examined whether the identified classes of functional interactions differed in their laminar distributions. Indeed, we found that different classes were differentially distributed across V1 layers such that one or more of the identified classes was often overrepresented among functional interactions within particular layers (*Figure 5—figure supplement 1*; $p<10^{-5}$). To simplify this result, we compared the proportion of CCGs in each class composed of two neurons in the same layer or different layers. We found that most of the asynchronous pairs were composed of neurons from different layers,

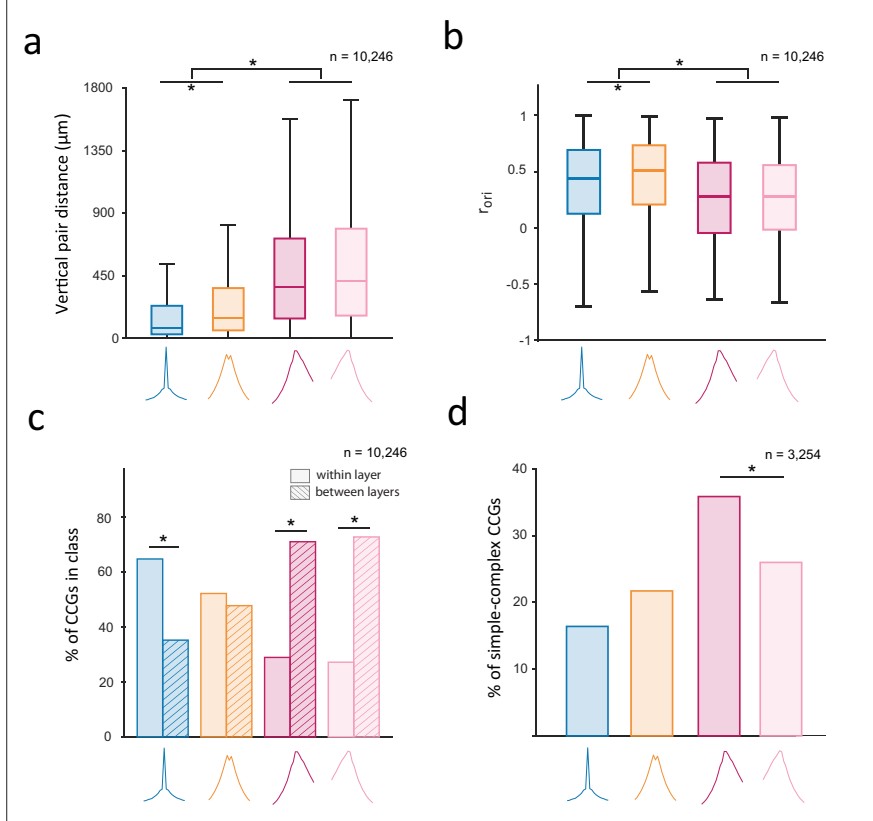

**Figure 5.** Corroboration of putative classes of functional interactions with V1 microcircuitry. (**a, b**) Boxplots of vertical pair distance (**a**) and orientation signal correlation (**b**) across the 4 identified CCG classes. Boxplots illustrate the medians, first and third quartiles, and non-outlier (1.5*IQR method) minima and maxima. Asterisks denote significant differences in medians between pairs of classes (Wilcoxon rank-sum test; $P<0.05$, Bonferroni corrected). (**c**) Percentage of CCGs in each class composed of two neurons from the same ('within layer') or different layers ('between layer'). In a-c, the reference neuron in a neuronal pair was selected randomly. (**d**) Distribution of putative CCG classes among neuronal pairs composed of a simple cell as the reference neuron and complex cell as the target neuron. Only the pairwise comparison between $F_{async}$ and $R_{async}$ is illustrated to show the direction of functional interactions between simple and complex cells.

The online version of this article includes the following figure supplement(s) for figure 5:

**Figure supplement 1.** Distribution of putative CCG classes across layer pairings.

while most of the synchronous pairs, particularly the $S_{sync}$ ones, were composed of neurons from the same layer (**Figure 5c**) (within vs between proportion: $S_{sync}$ [0.65 vs 0.35], $B_{sync}$ [0.52 vs 0.48], $F$ [0.29 vs 0.71], $R$ [0.27 vs 0.73], one proportion z-test: $S_{sync}$, $F$, $R$: $P<10^{-5}$; $B_{sync}$: p=0.022). This observation dovetails the relationship between pair distance and CCG class described above. Nonetheless, we found that cortical layer had an independent effect of distance on CCG class assignment among nearby (~200 μm) pairs of neurons. The location of neuronal pairs within the same or different, nearby layers predicted whether pairs belonged to the $B_{sync}$ class and the asynchronous classes ($F_{async}$ or $R_{async}$), but not the $S_{sync}$ class, when controlling for the effects of vertical distance (**Table 1**; logistic regression, $p<10^{-2}$). More specifically, CCGs composed of two neurons within the same layer had a higher probability of falling in the $B_{sync}$ class and a lower probability of falling into the asynchronous classes than CCGs with comparable vertical distances composed of two neurons in different cortical layers.

In addition to the laminar organization, V1 neurons exhibit clear differences in their receptive field properties. In particular, V1 neurons classically fall into two broad functional types: simple (S) and complex (C) cells (*De Valois et al., 1982*; *Hubel and Wiesel, 1962*; *Hubel and Wiesel, 1968*; *Movshon et al., 1978*; *Skottun et al., 1991*) (see also *Chance et al., 1999*; *Mechler and Ringach, 2002*; *Priebe et al., 2004*). Among the significant CCGs, a majority were comprised of pairs of complex cells (S/S=5.2%, C/C=63%; S/C=31.8%; one-proportion z-test: $p<10^{-5}$). Complex cells appear

**Table 1.** Dependence of putative classes on laminar pairing and vertical distance for pairs of neurons separated by 86–310 µm.

| Dependent variable (in/out of cluster) | Predictor | Coefficient | Standard error | p-Value |
|---|---|---|---|---|
| $S_{sync}$ | Distance | −0.0043 /µm | 0.001 | $4.00*10^{-5}$ |
| | Layer | 0.104/layer | 0.126 | 0.41 |
| $B_{sync}$ | Distance | −0.0017 /µm | 0.0007 | $9.24*10^{-3}$ |
| | Layer | 0.244/layer | 0.085 | $4.11*10^{-3}$ |
| $F_{async}$ or $R_{async}$ | Distance | 0.0032 /µm | 0.0005 | $4.23*10^{-7}$ |
| | Layer | −0.262/layer | 0.0802 | $1.13*10^{-3}$ |

Coefficients, standard errors, and p-values from logistic regressions predicting class membership using the distance between pairs of neurons and whether pairs were located in the same or different cortical layer(s). Only pairs of neurons with pair distances greater than the 5% of pairs located in different cortical layers (>86 µm) and less than 5% of pairs located in the same cortical layer (<310 µm) were included. Significant predictors are highlighted.

to receive converging input from groups of simple cells, and thus simple cells should lead rather than lag complex cells in their CCGs. To test this in our data, we compared the distribution of putative CCG classes among significantly correlated neuronal pairs composed of a simple cell as the reference neuron and a complex cell as the target neuron (*Figure 5d*). We found that the proportion of forward ($F_{async}$) CCGs was larger than the reverse ($R_{async}$) class ($p<10^{-5}$). Notably, although the dominant lead-lag relationship between simple and complex cells is consistent with established models of V1 (*Alonso and Martinez, 1998*; *Martinez and Alonso, 2001*; *Yu and Ferster, 2013*), there were also many CCG pairs in which complex cells led simple cells or where the pair fired synchronously. This heterogeneity in functional interactions between simple and complex cells is consistent with studies suggesting that simple and complex cells might arise from variations in a continuous process as opposed to being two clearly distinct populations (*Chance et al., 1999*; *Kim et al., 2021*; *Mechler and Ringach, 2002*; *Priebe et al., 2004*).

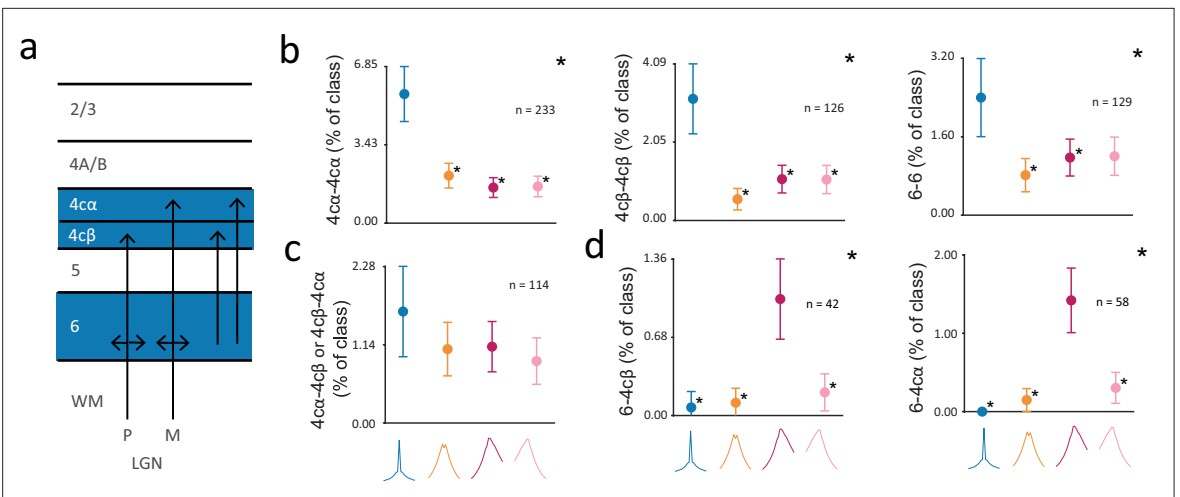

**Figure 6.** Distribution of different putative classes of functional interactions within V1 input layers. (**a**) Diagram of dLGN input to V1 layers 4cα, 4cβ, and 6; dLGN axons terminate in layers 4cα and 4cβ and layer 6, and layer 6 projects to layers 4cα and 4cβ. (**b**) Percentage of CCGs in each class composed of two neurons in layer pairings of 4cα-4cα, 4cβ-4cβ, or 6–6, out of all the pairwise layer pairing combinations. (**c**) Percentage of CCGs in each class composed of one neuron in layer 4cα and one in 4cβ (4cα-4cβ or 4cβ-4cα). In (**b-c**), the reference neuron in a neuronal pair was selected randomly. (**d**) Percentage of CCGs in each class composed of a reference neuron in layer 6 and a target neuron in 4cβ (left) or 4cα (right). For (**b-d**), error bars denote 95% confidence intervals. Large asterisks denote significant chi-squared test across all classes (p<0.05, Bonferonni corrected). Small asterisks denote significant chi-squared tests between a particular class and the class with the maximum percentage (p<0.05, Bonferroni corrected).

## Corroboration of different classes with V1 input and local circuitry

Previous studies have characterized the anatomical organization of dorsal lateral geniculate nucleus (dLGN) input to V1 in extensive detail (*Blasdel and Lund, 1983*; *Hendrickson et al., 1978*; *Hubel and Wiesel, 1972*). In the macaque brain, dLGN magnocellular and parvocellular axons primarily project to V1 layers 4cα and 4cβ, respectively, along with inputs that terminate in layer 6 (reviewed in *Briggs and Usrey, 2011*; *Lund, 1988*; *Merigan and Maunsell, 1993*; *Nassi and Callaway, 2009*; *Figure 6a*). However, the extent to which functional interactions within layers of V1 reflect these anatomical projections remains unclear. Thus, we examined the distribution of CCG classes across pairs of V1 input layers 4cα, 4cβ, and 6. We found that for the 4cα-4cα, 4cβ-4cβ, and 6–6 pairings, $S_{sync}$ CCGs were observed much more frequently than other CCG classes (*Figure 6b*) (chi-squared test; 4cα-4cα, 4cβ-4cβ: $P<10^{-5}$, 6–6: $P<10^{-2}$). This overrepresentation of $S_{sync}$ CCGs may reflect the fact that neurons in 4cα, 4cβ and 6 receive common and converging input from the dLGN. Furthermore, it is noteworthy that the $S_{sync}$ class was overrepresented in V1 input layers, but the $B_{sync}$ class was not.

In contrast to the overrepresentation of $S_{sync}$ CCGs within the input layers, this class of CCGs was not overrepresented in functional interactions between input layers. Of the four CCG classes, the proportions of each found among pairs composed of one neuron in layer 4cα and one neuron in layer 4cβ were statistically indistinguishable (*Figure 6c*) (chi-squared test; $p = 0.20$). The lack of an overrepresentation of $S_{sync}$ CCGs among 4cα/4cβ pairs could reflect the lack of synchrony between magnocellular (fast) and parvocellular (slow) inputs to V1. This result is noteworthy given that the average distance between neuronal pairs across 4cα and 4cβ was comparable to the distances between neuronal pairs within V1 input layer 6 (mean distance: 4cα/4cβ = 106 μm; 6/6=80 μm). In examining functional interactions between layers 4c and 6, we considered that a temporal offset between layer 6 and 4c neurons might exist given extensive projections from layer 6 pyramidal neurons to layer 4c (*Wiser and Callaway, 1996*). To test this, we examined the 6–4cα and 6-4cβ pairs in which the layer 6 neuron was the reference neuron in the crosscorrelation function. Indeed, in addition to observing that $S_{sync}$ CCGs were poorly represented, we found that the $F_{async}$ class was significantly overrepresented in these pairs (*Figure 6d*) (6–4cα: $p<10^{-5}$, 6-4cβ: $p<10^{-3}$).

Lastly, we examined the distribution of CCG classes across pairs of neurons involving layer 2/3 neurons. A wealth of evidence indicates that layer 2/3 neurons provide a major source of output to other neocortical areas (reviewed in *Callaway, 1998*; *Douglas and Martin, 2004*; *Felleman and Van Essen, 1991*; *Harris and Shepherd, 2015*; *Thomson and Lamy, 2007*). In macaque V1, layer 2/3 neurons send projections to higher visual areas such as V2 (*Livingstone and Hubel, 1984*; *Rockland, 1992*; *Sincich and Horton, 2005*) and V4 (*Yukie and Iwai, 1985*), and receive inputs from all the deeper cortical layers, including layers 4cα, 4cβ, 4A, 4B, 5, and 6 (*Blasdel et al., 1985*; *Callaway, 1998*; *Callaway and Wiser, 1996*; *Fitzpatrick et al., 1985*; *Kisvarday et al., 1989*; *Lachica et al.,*

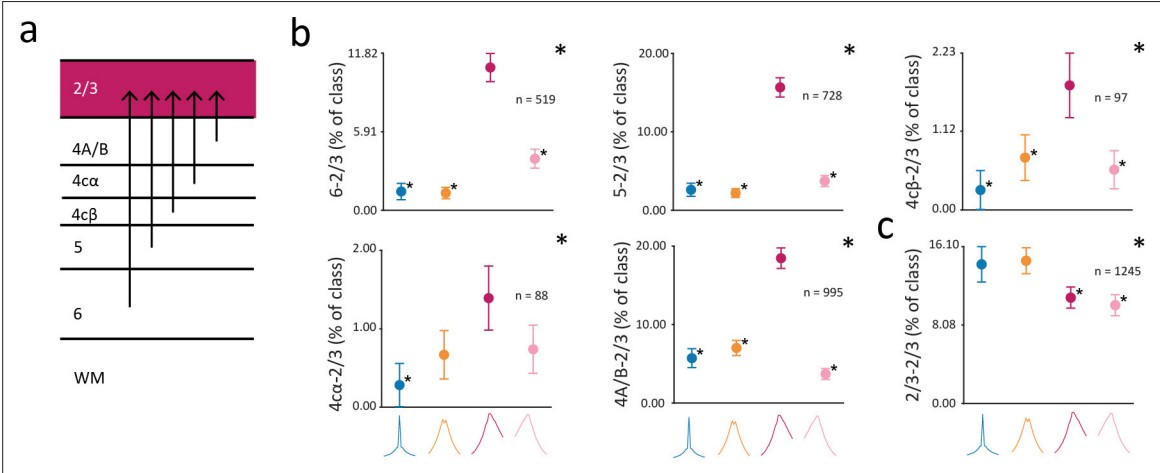

**Figure 7.** Distribution of different putative classes of functional interactions with layer 2/3. (**a**) Diagram of input to layer 2/3 from V1 laminar compartments. (**b**) Percentage of CCGs in each class composed of a reference neuron in layer 6, 5, 4cβ, 4cα, or 4A/B and a target neuron in layer 2/3. (**c**), Percentage of CCGs in each class composed of pairs of neurons in layer 2/3. The reference neuron in a neuronal pair was selected randomly. For b-c, conventions are the same as in *Figure 6*.

*1992*; *Lund and Boothe, 1975*; *Sawatari and Callaway, 2000*; *Vanni et al., 2020*; *Wiser and Callaway, 1996*; *Yarch et al., 2017*; *Yoshioka et al., 1994*; *Figure 7a*). Consequently, one might predict that a predominant proportion of projections to 2/3 neurons from other layers might be forward ones (*Callaway, 1998*; *Mejias et al., 2016*; *Schmidt et al., 2018*). Consistent with this prediction, we found that the forward ($F_{async}$) class was overrepresented among functional interactions from layers 6, 5, 4cβ, 4cα, and 4A/B to layer 2/3 (*Figure 7b*) (chi-squared test; 6: $p<10^{-5}$; 5: $p<10^{-5}$; 4cβ: $p<10^{-3}$; 4cα: $p<10^{-2}$; 4A/B: $p<10^{-5}$). In contrast, functional interactions within layer 2/3 exhibited a very different pattern. Within the same layer, the classes of 2/3-2/3 CCGs were more evenly represented, in stark contrast to the pattern of within-layer CCGs observed in the input layers (*Figure 6b*). Within layer 2/3, the $S_{sync}$ and $B_{sync}$ CCGs were overrepresented among functional interactions (*Figure 7c*) (chi-squared test: $p<10^{-4}$), and there was an equal representation of $S_{sync}$ and $B_{sync}$ CCGs among 2/3-2/3 pairings.

## Discussion

Using high-density recordings from single neocortical columns of macaque V1, we identified 1000s of functionally interacting neuronal pairs using established crosscorrelation approaches. The results demonstrate clear and systematic variations in the synchrony and strength of functional interactions within single V1 columns. Notably, we observed that in spite of residing within the same column, the functional interactions between pairs of V1 neurons depended heavily on their vertical distance within the column; both the peak lag and peak efficacy of CCGs between neuronal pairs changed dramatically within only a few hundred micrometers of vertical distance within the column. In addition, we found that the synchrony and strength of CCGs also depended on laminar location and the similarity of orientation tuning between neuronal pairs. We leveraged the statistical power provided by the large numbers of functionally interacting pairs to categorize functional interactions between neurons based on their crosscorrelation functions. These analyses identified distinct classes of functional interactions within the full population. Those distinct classes exhibited different distributions across defined laminar compartments, and those differences were consistent with known and/or expected properties of V1 cortical circuitry. The results demonstrate a novel utility of high-density neurophysiological recordings in assessing circuit-level interactions within local neuronal networks. Below, we discuss both the implications and the limitations of this approach.

### Effect of cortical distance on functional interactions

A wealth of previous evidence has established a clear effect of cortical distance on functional interactions, yet a majority of past studies have focused on the effect of horizontal distance across which large changes in shared input between neurons are expected. Evidence that spiking correlations and synchrony decline with horizontal cortical distance within V1 has been shown in cats (*Das and Gilbert, 1999*; *Gray et al., 1989*; *Hata et al., 1991*) (but see *Samonds et al., 2006*; *Schwarz and Bolz, 1991*), monkeys (*Chu et al., 2014*; *Krüger and Aiple, 1988*; *Maldonado et al., 2000*; *Smith and Kohn, 2008*), and in mice (*Denman and Contreras, 2014*). Very few studies have examined crosscorrelations among pairs of neurons within a single column, where the feedforward input is largely shared (e.g. *DeAngelis et al., 1999*). Longer timescale, spike count ('noise'), correlations, which have been widely assessed in studies of primate visual cortex (*Averbeck et al., 2006*) have been shown to be layer dependent within macaque V1 where weaker correlations occur in layer 4 (*Hansen et al., 2012*). However, no evidence that such correlations depend on distance independent of layer was observed. In contrast, measurement of crosscorrelations in earlier studies of V1 columns in cat indeed suggest that functional interactions are restricted to local regions across cortical depths (*Toyama et al., 1981b*). Within rat auditory cortex, functional interactions diminish dramatically within ~300 μm of vertical columnar distance (*Atencio and Schreiner, 2013*), similar to what is observed in rat somatosensory cortex (*Khateb et al., 2021*). A dependence of interactions on vertical distance is further supported by evidence from multiple whole-cell recordings in mouse visual cortex which demonstrates that connection probability decreases sharply within a distance of 250 μm (*Jiang et al., 2015*). Our observation that the peak efficacy of CCGs was greatly diminished within <200 μm within macaque V1 is thus consistent with estimates from other sensory cortices and species.

## Effect of orientation tuning similarity on functional interactions

Many previous studies have reported an effect of tuning similarity on functional interactions, yet most of these studies focused on interactions between pairs of neurons in different cortical columns. Evidence from horizontal array recordings in macaque V1 suggests that pairs of neurons selective for similar orientations (*Chu et al., 2014*; *Kohn and Smith, 2005*; *Smith and Kohn, 2008*), or color and luminance (*Chu et al., 2014*) tend to exhibit stronger functional interactions compared to pairs with dissimilar tuning, perhaps reflecting the functional specificity of horizontal connections (*Gilbert and Wiesel, 1989*). For example, one study found that the strength of functional interactions between neuronal pairs with the highest orientation signal correlations was nearly twice that of uncorrelated pairs (*Smith and Kohn, 2008*). Consistent with these previous studies, we also found that variations in tuning similarity resulted in robust differences in the synchrony and strength of correlated activity. However, our results demonstrate that an interdependence of tuning similarity and functional interaction exists even within single orientation columns. In addition, in contrast to comparisons across columns where tuning similarity appears to be the primary factor (*Chu et al., 2014*), we found that within single columns, vertical distance and orientation tuning similarity exerted distinct effects on the synchrony and strength of functional interactions.

**Table 2.** Number of CCGs with peaks or troughs significantly above or below noise.

| Stdev. above/ below noise | Number of pairs with peaks above noise | Number of pairs with troughs below noise |
|---|---|---|
| 1 | 33,502 | 15,582 |
| 2 | 33,502 | 11,347 |
| 3 | 31,660 | 4,537 |
| 4 | 25,180 | 1,555 |
| 5 | 18,739 | 603 |
| 6 | 13,757 | 265 |
| 7 | 10,246 | 136 |

The number of recorded pairs with a peak or trough at least 1–7 standard deviations (SD) above or below noise is shown. Only CCGs with peaks or troughs within 10ms of zero time lag were considered. 136 CCGs had troughs that were more than 7 SD below noise whereas 10,246 CCGs had peaks that were more than 7 SD above noise.

## Distinct classes of functional interactions

Upon clustering the full population of significant CCGs, we identified four putative classes of functional interactions. Notably, these classes of CCGs depicted the set of pairwise interactions that one might logically expect, namely two asynchronous classes (forward and reverse) and two synchronous ones (sharply and broadly synchronous). More importantly, the clustering-based evidence of distinct classes of CCGs was corroborated by the observation of highly differential distributions of those putative classes across cortical layers. For example, asynchronous CCG classes were more often observed among neuronal pairs within different layers, whereas synchronous pairs more often resided within the same layer. This corroboration of distinct classes extended to functional properties of V1 neurons as well in that simple cells were more often paired in a forward manner with complex cells, whereas the reverse was true for complex cells. Nevertheless, the existence of exactly four distinct classes among V1 pairs is by no means certain. Indeed, the choice of three classes was almost as valid as that of four in the clustering procedure (*Figure 4*). Yet, given the clear evidence of two asynchronous classes, and the differential distributions of broadly and sharply synchronous pairs across cortical layers, the choice of only three classes of CCGs seems less parsimonious than four. Although there appeared to be less evidence for the existence of five or more classes of functional interactions, that possibility cannot be ruled out either. For example, additional distinct classes of CCGs might be present, but significantly less frequent or weaker than the other four. Indeed, given their low incidence, our selection criteria already excluded CCGs with significant inhibitory peaks. As in previous studies, the frequency of excitatory CCGs in our dataset was considerably higher than that of inhibitory CCGs (*Aertsen and Gerstein, 1985*; *Hembrook-Short et al., 2019*; *Table 2*). Consequently, these additional classes were eliminated by the statistical threshold employed to identify significant CCGs. It is likely that additional distinct classes, excitatory or inhibitory, were also eliminated and/or simply fell within a mixture of the more dominant four classes that exceeded the statistical criterion. Future work will therefore be needed to more extensively characterize the distribution of distinct classes of spiking crosscorrelations among neuronal pairs in cortical columns.

## Classes of functional interactions and V1 microcircuitry

We found that the four putative classes of functional interactions were differentially distributed across the cortical column and across functional pairs of neurons. Most notably, the different CCG classes were observed in different proportions across V1 layers. In spite of those differences, it need not follow that the relative distribution of any specific putative class (e.g. sharply synchronous) fits with the known (or predicted) connectivity between different V1 neurons. For example, our observation that asynchronous CCGs ($F_{async}$ and $R_{async}$) were considerably more frequent among neuronal pairs situated in different laminae, and that synchronous CCGs ($S_{sync}$ and $B_{sync}$) were found among neurons in the same laminae, would not be expected if the differences in synchrony resulted primarily from measurement noise. Likewise, the observed disproportionality of $F_{async}$ and $R_{async}$ CCGs among functional interactions between simple and complex cells would not be expected if the two asynchronous classes were indistinguishable in our measurements. Instead, not only were the two classes disproportionate among simple and complex cells, but the overall direction of disproportionality was consistent with the known connectivity between the two functional classes of cells (*Alonso and Martinez, 1998*; *Martinez and Alonso, 2001*; *Yu and Ferster, 2013*). Thus, overall, we found that the pattern of differential distributions of the putative classes of CCGs across the column and across functional pairs of neurons was largely consistent with known properties of V1 microcircuitry.

## Functional interactions and synaptic connectivity

Analyses of the statistical dependencies between spike trains of two or more neurons have long played an important role in estimating how ensembles of neurons interact with one another (*Casile et al., 2021*; *Okatan et al., 2005*; *Perkel et al., 1967*). We interpret the functional interactions identified here in the same manner. Using statistical criteria employed in a recent study of mouse visual cortex (*Siegle et al., 2021*), we found that ~15% (10,246/68,579) of neuronal pairs within columns of macaque V1 exhibited significant functional interactions. The proportion of significant pairs is within the range observed in previous studies in macaque V1 (*Chu et al., 2014*; *Hembrook-Short et al., 2019*; *Kohn and Smith, 2005*; *Smith and Kohn, 2008*). This similarity with previous studies exists in spite of notable differences in the electrophysiological approach and statistical criteria. In addition, the proportion of significant pairs is also similar to previous CCG measurements made in the cat (*Alonso and Martinez, 1998*) and mouse V1 (*Denman and Contreras, 2014*; *Siegle et al., 2021*).

In addition, it has long been understood that the statistical interdependence of spike trains is not necessarily an indicator of synaptic connectivity among neurons (*Ostojic et al., 2009*; *Perkel et al., 1967*). Indeed, spike crosscorrelations are in no way a substitute for more direct measurements of synaptic connectivity, for example multi-patch recordings (*Cadwell et al., 2020*; *Jiang et al., 2015*; *Song et al., 2005*) or electron microscopy (*Bock et al., 2011*; *Lee et al., 2016*). Notably, only a subset of the identified CCG classes here are consistent with, although not necessarily indicative of, monosynaptically connected neuronal pairs. Specifically, pairs with non-zero time lags, such as the asynchronous ($F_{async}$ and $R_{async}$) classes we describe are those most likely to be monosynaptically connected. In our data, the asynchronous class amounted to ~9% of the total pairs tested. In contrast, the synchronous classes, particularly the $S_{sync}$ class, which are more likely to reflect pairs with common input (*Ostojic et al., 2009*; *Perkel et al., 1967*), amounted to ~6% of the total pairs tested. However, for the former group, it is uncertain of course how many of the total neuronal pairs represent genuine monosynaptic connections. Past studies have employed widely varying criteria to identify monosynaptic connections in a number of different brain structures and species (*Alonso and Martinez, 1998*; *English et al., 2017*; *Hembrook-Short et al., 2019*; *Liew et al., 2021*; *Reid and Alonso, 1995*; *Senzai et al., 2019*). Those criteria varied in the stringency with which CCGs are labeled as monosynaptic and include criteria that necessarily yield false negatives (*Senzai et al., 2019*).

Unfortunately, there exists little or no ground-truth measurements of the rate of synaptic connectivity, or common input, between neurons within single V1 columns in any species, though such measurements may be imminent (https://www.microns-explorer.org/). Nonetheless, the observed rate of asynchronous classes identified in our data is consistent with estimates of the connection probability among V1 neurons obtained in multi-patch, slice recordings. For example, recent studies identified synaptic connectivity in 5–12% of excitatory neurons in mouse/rat V1 (*Cadwell et al., 2020*; *Song et al., 2005*). Although estimates from such studies clearly underestimate the rate of connections given that many connections are cut in the slice preparation, they nonetheless provide a plausible

lower bound. Thus, in spite of the inherent arbitrariness of the statistical criteria employed in identifying significant CCGs, the observed proportion of asynchronous interactions among pairs in our data appears largely consistent with previous estimates. As a means of comparing the criteria used here to those of prior studies in monkey V1, we applied one criterion used in *Hembrook-Short et al., 2019* to our data, which required that CCG peaks be sharp and narrow (<5ms) to be considered monosynaptic. Using this criterion, we found that 86% of our asynchronous CCGs were categorized as consistent with monosynaptic connections. Thus, it seems likely that our asynchronous ($F_{async}$ and $R_{async}$) classes captured a reasonable proportion of monosynaptically connected neurons within the V1 column.

## Future studies

We found that the relative instances of different types of crosscorrelations observed among large populations of neuronal pairs may provide a means of constraining models of cortical microcircuits. This approach could prove particularly valuable in less experimentally tractable model systems such as nonhuman primates, or perhaps even in the human brain, where direct interrogation of microcircuits is difficult or not yet possible. In such cases, the ability of high-channel count, high-density, probes to dramatically increase the number of identifiable functional interactions within a local network of neurons is among their greater benefits. Our results thus far suggest that this approach works well and could be extended to examine higher-order interactions among larger sets of neurons, and to identify neuronal ensembles with distinct functional properties (*Fujisawa et al., 2008*; *Miller et al., 2014*; *See et al., 2018*). In addition, future studies should be able to compare local interactions across different putative cell types estimated from their spike waveforms (*Johnston et al., 2009*; *Lee et al., 2021*; *Mitchell et al., 2007*; *Wilson et al., 1994*) and/or spiking patterns (*Onorato et al., 2020*). Combined with measurements of functional interactions, such an approach could be used to constrain models of microcircuit architecture from neurophysiological data obtained from any number of uniquely evolved primate brain structures.

## Methods

### Experimental model and subject details

Anesthetized recordings were conducted in 2 adult male rhesus macaques (*Macaca Mulatta*, M1, 13 kg; M2, 8 kg). All experimental procedures were in accordance with National Institutes of Health Guide for the Care and Use of Laboratory Animals, the Society for Neuroscience Guidelines and Policies, and with approved Institutional Animal Care and Use Committee (IACUC) protocol (#APLAC-9900) of Stanford University.

### Electrophysiological recordings

Prior to each recording session, treatment with dexamethasone phosphate (2 mg per 24 hr) was instituted 24 hr to reduce cerebral edema. After administration of ketamine HCl (10 mg/kg body weight, intramuscularly), monkeys were ventilated with 0.5% isoflurane in a 1:1 mixture of $N_2O$ and $O_2$ to maintain general anesthesia. Electrocardiogram, respiratory rate, body temperature, blood oxygenation, end-tidal $CO_2$, urine output and inspired/expired concentrations of anesthetic gases were monitored continuously. Normal saline was given intravenously at a variable rate to maintain adequate urine output. After a cycloplegic agent was administered, the eyes were focused with contact lenses on a CRT monitor. Vecuronium bromide (60 µg/kg/hr) was infused to prevent eye movements.

 With the anesthetized monkey in the stereotaxic frame, an occipital craniotomy was performed over the opercular surface of V1. The dura was reflected to expose a small (~3 mm$^2$) patch of cortex. Next, a region relatively devoid of large surface vessels was selected for implantation, and the Neuropixels probe was inserted with the aid of a surgical microscope. Given the width of the probe (70 µm x 20 µm), insertion of it into the cortex sometimes required multiple attempts if it flexed upon contacting the pia. The junction of the probe tip and the pia could be visualized via the (Zeiss) surgical scope and the relaxation of pia dimpling was used to indicate penetration, after which the probe was lowered at least 3–4 mm. Prior to probe insertion, it was dipped in a solution of the DiI derivative FM1-43FX (Molecular Probes, Inc) for subsequent histological visualization of the electrode track.

 Given the length of the probe (1 cm), and the complete distribution of electrode contacts throughout its length, recordings could be made either in the opercular surface cortex (M1) or within

the underlying calcarine sulcus (M2), by selecting a subset of contiguous set of active contacts (n=384) from the total number (n=986). Receptive fields (RFs) from online multi-unit activity were localized on the display using at least one eye. RF eccentricities were ~4–6° (M1) and ~6–10° (M2). Recordings were made at 1–3 sites in one hemisphere of each monkey. At the end of the experiment, monkeys were euthanized with pentobarbital (150 mg kg⁻¹) and perfused with normal saline followed by 1 liter of 1% (wt/vol) paraformaldehyde in 0.1 M phosphate buffer, pH 7.4.

## Visual stimulation

Visual stimuli were presented on a LCD monitor NEC-4010 (Dimensions = 88.5 (H)* 49.7 (V) cm, pixels = 1360 * 768, frame rate = 60 Hz) positioned 114 cm from the monkey. Stimuli consisted of circular drifting Gabor gratings (2 deg./sec., 100% Michelson contrast) positioned within the joint RFs of recorded neurons monocularly. Gratings drifted in 36 different directions between 0–360° in 10° steps in a pseudorandom order. Four spatial frequencies (0.5, 1, 2, 4 cycle/deg.) were tested and optimal SFs were determined offline to categorize V1 neurons into simple or complex cell. The stimulus in each condition was presented for 1 second and repeated 5 or 10 times. A blank screen with equal luminance to the Gabor patch was presented for 0.25 s during the stimulus interval.

## Data acquisition and spike sorting

Raw spike-band data was sampled and recorded at 30 kHz. It was then median-subtracted and high-pass filtered at 300 Hz during the pre-processing stage. Spike-sorting was carried out with Kilosort2 (https://github.com/MouseLand/Kilosort; *Pachitariu et al., 2022*) to find spike times and assign each spike to different units. The raw sorted data was then manually curated in Phy (https://github.com/cortex-lab/phy; *Rossant et al., 2022*) to remove spikes with atypical waveforms and perform minimal merging and splitting. One potential issue with the template-matching approach used by Kilosort2 is that the algorithm will occasionally fit a new template from the residual after subtracting the first template from the original data. This artificial template/neuron will share an abnormal number of double-counted spikes with the real neuron, resulting in a high zero-time lag synchrony between those two neurons. To examine whether this issue may affect our results, we used a criteria suggested by a previous study (*Siegle et al., 2021*) to identify double-counted spikes by counting spikes with peak times within 5 samples (0.167ms) and from pairs of neurons within 50 μm (~5 channels). Out of the total 68,579 pairs of neurons included in this current study, we found that only 7 pairs shared more than 20% overlapping spikes. Considering that we identified 1408 pairs of neurons in the sharply synchronous class, those potentially artificial pairs contributed to less than 0.5% of this class. Although we did not remove those pairs from the data, we believe their contributions are negligible. Here, we list key parameters in Kilosort2 that may affect the 'double-counting': Ops.th=[10,4]; Ops.lam=20; Ops.AUCsplit=0.9; Ops.ThPre=8; Ops.spkTh=-6. Moreover, only neurons with a minimum firing rate of 3 spikes/s were included in the study.

## Layer assignment

The laminar location of our recording sites was estimated based on a combination of functional analysis and histology results. For each recording, we first performed the current source density (CSD) analysis on the stimulus-triggered average of local field potentials (LFP). LFP were low-pass filtered at 200 Hz and recorded at 2500 Hz. LFP signals recorded from each four neighboring channels were averaged and realigned to the onset of visual stimulus. CSD was estimated as the second-order derivatives of signals along the probe axis using the common five-point formula (*Nicholson and Freeman, 1975*). The result was then smoothed across space (σ=120 μm) to reduce the artifact caused by varied electrode impedance. We located the lower boundary of the major sink (the reversal point of sink and source) as the border between layer 4c and layer 5/6. Based on this anchor point, we assign other laminar compartment borders using the histological estimates.

## Single neuron properties

To characterize the visual properties of each neuron, the stimulus evoked activity was assessed using mean firing rate (spikes/s) over the entire stimulus presentation period, offset by a response latency of 30ms. Only responses to the preferred spatial frequency were used. Modulation ratio was defined as F1/F0, where F1 and F0 are the amplitude of the first harmonic at the temporal frequency of drifting

grating and constant component of the Fourier spectrum to the neuron's response to preferred orientation. Simple cells were defined as cells with modulation ratio larger than 1, and complex cells have modulation ratios smaller than 1 (*De Valois et al., 1982*; *Skottun et al., 1991*).

## Signal correlations

To measure the similarity of orientation tuning between neuronal pairs, we computed an orientation signal correlation ($r_{ori}$). The orientation signal correlation was defined as the Pearson's correlation coefficient between the mean responses of two neurons to each of the 36 stimulus orientations (*Smith and Kohn, 2008*). For each neuron and orientation, a single mean response was computed by averaging spiking activity over the entire duration of stimulus presentation (1 s) across all trials with a particular orientation.

## Cross-correlograms (CCGs)

To measure correlated firing, we computed the crosscorrelation between spike trains of all pairs of simultaneously recorded neurons (*Jia et al., 2013*; *Siegle et al., 2021*; *Smith and Kohn, 2008*; *Zandvakili and Kohn, 2015*). We focused on the spiking activity within the 0.4–1 s window of each visual stimulus presentation, which ensured that the analysis was not affected by the transient response to stimulus onset. To mitigate firing rate effects, we normalized the cross-correlation for each pair of neurons by the geometric mean of their firing rates. Thus, the cross-correlogram ($CCG$) for a pair of neurons ($j, k$) was defined as follows:

$$CCG\left(\tau\right)_{j-k} = \frac{\frac{1}{M}\sum_{i=1}^{M}\sum_{t=1}^{N-\tau} x_j^i\left(t\right) \times x_k^i\left(t+\tau\right)}{\theta\left(\tau\right)\sqrt{\lambda_j \lambda_k}}$$

where $M$ is the number of trials, $N$ is the number of time bins within a trial, $\tau$ is the time lag, $x_j^i\left(t\right)$ is one if neuron $j$ fired in time bin $t$ of trial $i$ and zero otherwise, and $\lambda_j$ is the mean firing rate of neuron j computed over the same bins used to compute the CCG at each time lag. $\theta\left(\tau\right)$ is a triangular function, $\theta\left(\tau\right) = N - |\tau|$, that corrects for the difference in the number of overlapping bins at different time lags. We denote the CCG computed with neuron j as the first (reference) neuron and k as the second (target) neuron in the correlation function as j-k.

To correct for correlation due to stimulus-locking or slow fluctuations in population response (e.g. gamma-band activity), we computed a jitter-corrected cross-correlogram by subtracting a jittered cross-correlogram from the original cross-correlogram:

$$CCG_{corrected} = CCG_{original} - CCG_{jittered}$$

The jittered cross-correlogram ($CCG_{jittered}$) reflects the expected value of cross-correlograms computed from all possible jitters of each spike train within a given jitter window (*Harrison and Geman, 2009*; *Smith and Kohn, 2008*). The jittered spike train preserves both the PSTH of the original spike train across trials and the spike count in the jitter window within each trial. As a result, jitter correction removes the correlation between PSTHs (stimulus-locking) and correlation on timescales longer than the jitter window (slow population correlations). Here, a 25 ms jitter window was chosen based on previous studies (*Jia et al., 2013*; *Siegle et al., 2021*; *Zandvakili and Kohn, 2015*).

We classified a CCG as significant if the peak of the jitter-corrected CCG occurred within 10ms of zero and was more than seven standard deviations above the mean of the noise distribution. The noise distribution for a CCG was defined as the flanks of the jittered-corrected CCG ($\{CCG\left(\tau\right) \mid 100 \geq |\tau| \geq 50\}$). This significance criterion was chosen based on that of *Siegle et al., 2021*. All analyses presented here involve only significant, jitter-corrected cross-correlograms. Note that the criterion identifies only positive peaks in the CCG and excludes significant inhibitory correlations. However, consistent with earlier studies (*Aertsen and Gerstein, 1985*; *Hembrook-Short et al., 2019*), we found that the frequency of CCGs with significant troughs was approximately 40x lower than those with significant peaks (*Table 2*).

## Classification of Cross-correlogram

To identify distinct classes of cross-correlation functions, we clustered significant crosscorrelations. We only analyzed crosscorrelation functions between $\tau = -10$ and $\tau = 10$ such that our input CCGs had 21 features, corresponding to the 21 crosscorrelation values between $\tau = -10$ and $\tau = 10$. For

clustering, we included two crosscorrelation functions for each pair of neurons $(j, k)$, one computed using the above CCG function with neuron j as the reference neuron $(j - k)$ and the other with neuron k as the reference neuron $(k - j)$. This was done in order to avoid introducing biases in the direction of the CCG templates. We z-scored each CCG prior to clustering to encourage clustering based on the shape of the correlation function rather than its magnitude. For subsequent statistical analyses, only a single neuron in the pair was used as the reference.

To simplify the clustering problem, we used t-distributed stochastic neighbor embedding (t-SNE) to reduce our input data with 21 features to 3 features (tsne, MATLAB R2019a). t-SNE was used instead of principal component analysis (PCA) because it is more robust to outliers since it captures neighbor relationships in the input space. We clustered the dimensionality-reduced data using k-means with $k = 1$ to 10 (50 replicates, 100 max iterations, kmeans MATLAB R2019a). To determine the optimal number of clusters, we used two complementary approaches, the elbow method and silhouette method. The elbow method selects $k$ based on the magnitude of the change in the variance explained by clustering as $k$ increases. For a set of points $S = \{s_1, s_2, \ldots, s_n\}$ divided into $k$ clusters $S_1, S_2, \ldots, S_k$, the percent of variance explained by clustering $(\eta_k)$ is:

$$\eta_k = \frac{(TSS - WCSS_k)}{TSS}$$

$$TSS = \sum_{x \in S} \|x - mean(S)\|$$

$$WCSS_k = \sum_{i=1}^{k} \sum_{x \in S_i} \|x - mean(S_i)\|$$

where $TSS$ denotes the total sums of squares and $WCSS_k$ denotes the sum of within cluster sums of squares over all clusters. The optimal number of clusters occurs at the point where the percent of explained variance plateaus (or 'elbows') as the number of clusters increases. The silhouette criterion captures how similar a point is to its own cluster versus how different it is from the nearest cluster that it is not a member of. We computed the silhouette criterion using MATLAB's 'evalclusters' function with default parameters (MATLAB R2019a).

## Statistical analyses

The effects of vertical pair distance and orientation signal correlation on CCG peak lag and peak efficacy were fit using linear and exponential functions. In linear regressions predicting CCG peak lag, all significant CCGs were included, and mean squared error was used as the cost function for regressions. In linear and exponential regressions predicting CCG peak efficacy, only significant CCGs with non-outlier peaks (1.5*IQR criterion) were included, and mean absolute error was used as the cost function for regressions to encourage fit of the plotted median peak efficacies.

The relationships between classes of functional interactions and signal correlation/pair distance were evaluated using Wilcoxon rank-sum tests, and the relationship between functional class and layer/cell type pairings was assessed using chi-squared tests or one proportion z-tests. Finally, the dependence of functional class on whether a CCG was composed of two neurons within the same or different cortical layer(s) with comparable vertical distance was assessed using logistic regression.

## Distance matching

Distance matching was used to compare the effects of orientation signal correlation on CCG peak lag and peak efficacy among neuronal pairs with comparable cortical distances. To match pairs with comparable distances (*Figure 3C*), we sorted significant CCGs by cortical distance, then paired the CCGs with the smallest and second smallest distances and paired the CCGs with the third and fourth smallest distances and so forth. Thus, every significant CCG was paired with exactly one other significant CCG, resulting in 5122 pairs. To verify that this procedure effectively matched pairs of CCGs with comparable cortical distance, we examined the difference in cortical distance for distance-matched pairs. More than 99% (5067/5122) of the distance-matched pairs had a difference in cortical distance of less than 2 μm. Finally, we examined the correlation between the difference in CCG peak lag

or peak efficacy and difference in signal correlation for matched pairs to determine whether signal correlation predicts peak lag or peak efficacy when controlling for distance.

## Acknowledgements

We thank Jonathan C Horton for extensive help with the recordings and histology. We thank Tim Harris and Karel Svoboda for providing the Neuropixels probes, Shellie Hyde and Sam Baker for technical assistance.

## Additional information

### Competing interests

Tirin Moore: Senior editor, *eLife*. The other authors declare that no competing interests exist.

### Funding

| Funder | Grant reference number | Author |
|---|---|---|
| National Institute of Neurological Disorders and Stroke | NS116623 | Tirin Moore |
| National Eye Institute | EY014924 | Tirin Moore |
| National Eye Institute | EY029759 | Xiaomo Chen |

The funders had no role in study design, data collection and interpretation, or the decision to submit the work for publication.

### Author contributions

Ethan B Trepka, Conceptualization, Data curation, Software, Formal analysis, Supervision, Validation, Investigation, Visualization, Methodology, Writing – original draft, Project administration, Writing – review and editing; Shude Zhu, Conceptualization, Resources, Data curation, Software, Formal analysis, Supervision, Validation, Investigation, Visualization, Methodology, Writing – original draft, Project administration, Writing – review and editing; Ruobing Xia, Conceptualization, Software, Formal analysis, Investigation, Visualization, Methodology, Writing – review and editing; Xiaomo Chen, Resources, Data curation, Funding acquisition, Investigation, Writing – review and editing; Tirin Moore, Conceptualization, Supervision, Funding acquisition, Investigation, Visualization, Methodology, Writing – original draft, Project administration, Writing – review and editing

### Author ORCIDs

Ethan B Trepka (iD) http://orcid.org/0000-0002-3002-800X
Shude Zhu (iD) http://orcid.org/0000-0002-8674-9607
Tirin Moore (iD) http://orcid.org/0000-0002-3345-2930

### Ethics

All experimental procedures were in accordance with National Institutes of Health Guide for the Care and Use of Laboratory Animals, the Society for Neuroscience Guidelines and Policies, and with approved Institutional Animal Care and Use Committee (IACUC) protocol (#APLAC-9900) of Stanford University.

### Decision letter and Author response

Decision letter https://doi.org/10.7554/eLife.79322.sa1
Author response https://doi.org/10.7554/eLife.79322.sa2

## Additional files

### Supplementary files
• MDAR checklist

## Data availability

All the raw data generated as part of this study are publicly accessible. All the raw code generated for analyzing the data has already been deposited to GitHub and is currently freely accessible (https://github.com/et22/functional_connections_macaque_v1, copy archived at swh:1:rev:39dda9dffab8f6e54069fb6514d2230200412472).

The following dataset was generated:

| Author(s) | Year | Dataset title | Dataset URL | Database and Identifier |
|---|---|---|---|---|
| Trepka E, Zhu S, Xia R, Chen X, Moore T | 2022 | Functional Interactions Among Neurons within Single Columns of Macaque V1 | https://dx.doi.org/10.5061/dryad.x3ffbg7p2 | Dryad Digital Repository, 10.5061/dryad.x3ffbg7p2 |

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
