## [Editor Report]

This is an important paper which shows how high-density neurophysiological recordings in non-human-primates can be used to identify inter-neuronal interactions based on cross-correlations. This provides valuable insights such as the dependence of correlations on vertical distance and orientation tuning. Overall the techniques used here are compelling and set a standard for recordings in non-human-primates. The paper is of interest for a broad audience of neuroscientists that performs electrophysiological recordings or is interested in functional interactions among neuron pairs.

---

## [Decision Letter]

**Decision letter after peer review:**

Thank you for submitting your article "Functional Connections Among Neurons within Single Columns of Macaque V1" for consideration by *eLife*. Your article has been reviewed by 2 peer reviewers, and the evaluation has been overseen by a Reviewing Editor and Timothy Behrens as the Senior Editor. The following individual involved in the review of your submission has agreed to reveal their identity: Jens Kremkow (Reviewer #1).

Essential revisions:

1. Although the paper does provide rich information on interactions within local cortical circuits, the main weakness of the paper is using the term "functional connection" in an imprecise manner. Cross-correlograms (CCG) of spike trains of pairs of neurons show different shapes depending on the underlying connectivity and not all significant peaks in CCGs reflect functionally connected neuron pairs. For example, CCGs of synaptically connected neuron pairs show a transient peak that is offset from the 0-ms lag due to the synaptic delay. CCGs with this shape thus reflect "functionally connected neuron pairs". In contrast, common inputs to pairs of neurons can induce significant peaks in CCGs, despite the fact that these neurons are only correlated but not functionally connected (e.g. Ostojic et al. 2009). Therefore, taking the shape of significant CCGs into account is important when discussing "functionally connected neuron pairs". While the authors mention this point in the paper, the term "functional connection" is nonetheless used irrespective of the CCG shapes which can be confusing to the reader. Moreover, the authors claim that the method allows identifying "1000s of functionally connected neuronal pairs". This statement is likely not fully supported by the data, evident by the fact that CCGs with the shape of mono-synaptic connections (transient and non-zero lag peak) are not among the distinct classes of CCGs shown in Figure 4.

The term "functionally connection" implies a synaptic connection between a pair of neurons. However, the majority of CCGs shown in the paper likely reflect correlated activity at fast timescales rather than true functionally connections. Because this can be confusing to readers I recommend defining the terminology more precisely and also using the term "functionally connected neuron pairs" only in cases where this is justified. This is particularly important because one claim of the paper is that Neuropixels probes allow identifying "1000s of functionally connected neuronal pairs". Please either show that indeed this large number of synaptically connected neurons is in the dataset, or change the title/abstract to match the results.

2. Likewise, it is surprising that CCGs that reflect mono-synaptic connections are not among the distinct classes shown in Figure 4? Why is that? Given the large number of tested interactions, and the claims of the paper, we would have expected that mono-synaptic connections form one of the distinct classes. It could be interesting and useful to specifically identify and study mono-synaptic connections, e.g. by employing methods reported in Liew et al. 2021 JNP, or by other approaches.

3. It is well established that fast-spiking neurons in the cortex receive stronger inputs from neurons in the local circuitry as compared to regular spiking neurons. Because fast-spiking and regular spiking neurons can be distinguished based on their spike waveform in extracellular recordings it could be interesting, but not required, to see whether this cell-type dependent connection strength is also evident in this dataset. This could add to the significance of the work and provide another angle to investigate circuit interactions. Again, this is only a suggestion.

4. It would be useful to show the CCGs in Figure 1e in a higher temporal resolution around the peak such that the lag of the peak is visible.

5. It seems odd to treat the forward and reverse correlation functions as two distinct types since the labeling is arbitrary. That is, swapping the label for the reference neuron and its partner would flip the correlation function from one category to the other. Since the choice of the reference neuron is arbitrary, it is not clear why there are two categories. Please provide justification.

6. Separate from this issue, the labeling looks erroneous to the best of our knowledge. That is, 'forward' correlation functions have more mass at negative lags. This is opposite to the conventional definition. Mass at negative lags means the partner neuron tends to fire before the reference neuron, which is not a forward connection (reference driving partner) but the opposite. Please clarify.

7. Please provide the species used.

8. Page 10: It was not obvious that the regression coefficients could be directly compared since the covariates (pair distance and r_ori) have very different magnitudes. Please clarify.

9. Please provide an explanation of how the spike sorting was done. One concern about the sharp correlation peaks is that they are artifactual, resulting from issues in Kilosort (https://github.com/MouseLand/Kilosort/issues/29). In brief, Kilosort can match the same spike waveform to two different templates, one template capturing most of the waveform shape and the second capturing the residual. This issue is discussed on the Github Kilosort page. Are the authors sure that this does not contribute to/drive the sharp peak phenomenology? One way to provide reassurance would be to provide more information on the shape and width of the sharp peaks. If they are ~1 ms wide, it seems more likely to be artifactual than biological.

10. Please provide the proportion of simple and complex neurons in the data set.

11. Discussion: The authors do a good job of discussing the limitations of correlation analysis for identifying synaptic connections. However, they refer to their R and F shapes (9% of cases) as potentially capturing such connections. Traditionally (e.g. Reid and Alonso), correlation function consistent with synaptic connections must pass a more stringent description than a simple asymmetry with respect to zero time lag. Namely, they need to have a sharp peak clearly offset from zero. This difference in stringency should be incorporated into the Discussion if the authors wish to propose that R and F types are related to connectivity.

12. Line 635: The definition of the correlation function seems different from previous work. The correlation function appears normalized by a quantity that depends on the time lag of one neuron and not the other (the denominator). Please clarify the notation.

13. Figure 2B: There is an oscillation in the marginal histogram of peak delays. This is also evident in Figure 3C. Please clarify.

---

## [Author Response]

Essential revisions:1. Although the paper does provide rich information on interactions within local cortical circuits, the main weakness of the paper is using the term "functional connection" in an imprecise manner. Cross-correlograms (CCG) of spike trains of pairs of neurons show different shapes depending on the underlying connectivity and not all significant peaks in CCGs reflect functionally connected neuron pairs. For example, CCGs of synaptically connected neuron pairs show a transient peak that is offset from the 0-ms lag due to the synaptic delay. CCGs with this shape thus reflect "functionally connected neuron pairs". In contrast, common inputs to pairs of neurons can induce significant peaks in CCGs, despite the fact that these neurons are only correlated but not functionally connected (e.g. Ostojic et al. 2009). Therefore, taking the shape of significant CCGs into account is important when discussing "functionally connected neuron pairs". While the authors mention this point in the paper, the term "functional connection" is nonetheless used irrespective of the CCG shapes which can be confusing to the reader. Moreover, the authors claim that the method allows identifying "1000s of functionally connected neuronal pairs". This statement is likely not fully supported by the data, evident by the fact that CCGs with the shape of mono-synaptic connections (transient and non-zero lag peak) are not among the distinct classes of CCGs shown in Figure 4.The term "functionally connection" implies a synaptic connection between a pair of neurons. However, the majority of CCGs shown in the paper likely reflect correlated activity at fast timescales rather than true functionally connections. Because this can be confusing to readers I recommend defining the terminology more precisely and also using the term "functionally connected neuron pairs" only in cases where this is justified. This is particularly important because one claim of the paper is that Neuropixels probes allow identifying "1000s of functionally connected neuronal pairs". Please either show that indeed this large number of synaptically connected neurons is in the dataset, or change the title/abstract to match the results.

We thank the reviewer for raising this point as we agree that it is important to use the correct terminology. In particular, as with any terminology, we think it is crucial to use terms as they are used in the literature. As we did not invent the term “functional connectivity”, it is thus important to consider how it has been used previously. The reviewer states that not all significant CCGs reflect “functionally connected” pairs thus implying that the phrase “functionally connected” must be synonymous with “synaptically connected”, or “mono-synaptically connected”. However, this is not how the term has generally been used in the literature. For example, Aertsen and Gerstein (1985) noted that: “The commonly used framework for investigating connectivity in a neural network is the cross-correlation function of simultaneously recorded spike trains. The presence of significant departures from a flat background is interpreted as indicative of a functional connection, either direct (through one or more synapses) or indirect (shared input, either neural or stimulus-induced)”. (Gerstein of course was one of the pioneers of the use and interpretation of CCGs.) Similarly, Friston and colleagues define “functional connectivity” as “statistical dependencies among remote neurophysiological events” (Friston, 1994, 2011).

In addition, many other studies have used the term “functional connectivity” similarly to the way we have and have treated CCGs consistent with common input as a type of “functional connection”. These studies include: Toyama, 1988; Schwarz and Bolz, 1991; Menz and Freeman, 2004; Denman and Contreras, 2014; Dann et al., 2016. In Menz and Freeman (2004), the authors define three types of functional connections:

“monosynaptic connection”, “polysynaptic connection” and “common input connection”. Denman and Contreras (2014), used the term “functional connectivity” throughout their paper and differentiated two types of functional connectivity: zeros-spanning and offset synchrony, similar to our “synchronous” and “asynchronous” categories. Aertsen and colleagues (1989) use the term “effective connectivity” and explicitly argue that “…The notion of ‘effective connectivity’ usually can be split into two types of circuitry: direct interactions and shared input.” A similar argument is made by Constantinidis et al., (2001).

In the above sense, “functional connectivity” is statistical in nature and refers to the temporal correlations between time series from different neuronal spike trains. This sense of the term avoids conclusions about the presence or absence of an underlying causal effect, which strictly speaking cannot be inferred from *any* CCG as it is not a ground truth measure. In the logical and grammatical sense, the word “functional” in “functional connectivity” simply qualifies the word “connection”, since by itself, the word “connection” must mean a synaptic one. In other words, in the literature, the term was used by investigators to distinguish statistically determined “connectivity” from synaptic connectivity, the latter of which must be directly determined by other means. Thus, our use of the term “functional connectivity” neither contradicts prior uses and definitions in the literature nor is it illogical or imprecise, and we are by no means the first to include CCGs consistent with common input as a type of “functional connection”. Indeed, this latter type of CCG is particularly important in our study as it appears to validate the expectation of common LGN input to neurons within particular thalamorecipient compartments (Figure. 6a-b).

There are indeed some prior studies that use the term “functional connectivity” to imply (mono)synaptic connections, but they are relatively few in number. For example, the term has been used this way to describe CCGs between simple and complex cells (Alonso and Martinez, 1998), thalamocortical connections (Bereshpolova, Hei, Alonso and Swadlow, 2020) and retinocollicular connections (Sibille et al., 2021). However, notably, the term itself is not clearly defined anywhere in these papers, and so the precise meaning, and its correspondence with uses in other studies, is left ambiguous. The reviewer cites Ostojic et al., 2009 to argue that neuronal pairs with significant CCGs induced by common input are only correlated but cannot be called “functionally connected”. However, Ostojic et al. do not use or define the term “functional connection” or “functional connectivity” at all in their paper. Instead, they use the term “functional interaction” and state in their introduction that “…statistically significant cross-correlations arise from the presence of a direct synaptic connection and/or from common or correlated inputs to the two neurons”.

To summarize, we agree that accurate use of terminology is important. Indeed, our use of “functional connectivity” is consistent with its use in the prior literature. Moreover, in the original manuscript, we had explicitly defined our use of the term in the discussion, and clarified which subset of identified CCG classes *might* be consistent with mono-synaptically connected pairs. Nevertheless, we also agree that it may be more important to provide the clearest possible use of terminology, given that many readers will be less familiar with prior terms used in the literature. Thus, we have expanded the introduction of the revised paper to both summarize past interpretations of CCGs and the terminology used in the literature and to clarify upfront our own interpretations and terminology to minimize confusion. In addition, to further avoid confusion regarding our interpretation of CCGs generally, we have adopted the term “functional interaction” as an alternative, since it avoids the word “connectivity” altogether. In the revised paper, we address the extent to which particular subclasses of significant CCGs are, or are not, consistent with monosynaptic connections in their respective instances.

2. Likewise, it is surprising that CCGs that reflect mono-synaptic connections are not among the distinct classes shown in Figure 4? Why is that? Given the large number of tested interactions, and the claims of the paper, we would have expected that mono-synaptic connections form one of the distinct classes. It could be interesting and useful to specifically identify and study mono-synaptic connections, e.g. by employing methods reported in Liew et al. 2021 JNP, or by other approaches.

First, it is important to emphasize that the distinct classes shown in Figure 4 are templates of CCG clusters (i.e. not individual CCGs) and that it is not necessarily the case that we should expect any of those templates to be monosynaptic-looking. Correlations between pairs of neurons within cortical columns likely reflect more than a single form of interaction, e.g. monosynaptic connection plus common inputs, or mono and disynaptic interactions, etc. Thus, when clustering CCGs, the dominant shapes of the correlations captured in the CCG templates (from large numbers of individual CCGs) likely reflect those mixtures. Unlike simple circuits, such as thalamocortical ones where one expects a single dominant, monosynaptic interaction between neuronal pairs (i.e., thalamic principal cell to layer IV cell), intracortical circuits are considerably more complex, and thus it may be overly simplistic to assume that one dominant, and singularly monosynaptic-looking CCG shape should emerge from the clustering.

Second, the presumption that mono-synaptic-looking connections are not part of our asynchronous cluster is wrong. When we apply the Liew et al. criteria (peak lag of 1-4 ms, peak > 3.5 s.d. trial-shuffled noise) to the asynchronous CCGs, those considered to be monosynaptic connections comprise 63% of the total group. Moreover, when we apply an alternative criterion to capture CCGs with sharp peaks (Hembrook-Short et al.) to the asynchronous CCGs, those considered to have sharp peaks consistent with monosynaptic connectivity comprise 86% of the total group. Thus, if we assume that the applied criteria are correct, a majority of the asynchronous class of CCGs may consist of a single, monosynaptic interaction between neuronal pairs. Note also that if we average across a large set of monosynaptic CCGs with varying peak lags (i.e. 1-4ms), the resultant template will necessarily have a broader peak than individual CCGs. Moreover, it is important to note that monosynaptic CCGs likely look quite different in different cortical regions, species, and in thalamocortical versus cortico-cortical connections. The study cited by the reviewer (Liew et al. 2021), for example, addressed monosynaptic connections between VPM and barrel cortex in rodents. The time lag and complexity of cortico-cortical interactions in macaque V1 likely differ substantially from that of thalamocortical connections in rodent barrel cortex (Stratford et al. 1996; Gil, Connors, Amitai, 1999; Gilman, Medalla, Luebke, 2017).

3. It is well established that fast-spiking neurons in the cortex receive stronger inputs from neurons in the local circuitry as compared to regular spiking neurons. Because fast-spiking and regular spiking neurons can be distinguished based on their spike waveform in extracellular recordings it could be interesting, but not required, to see whether this cell-type dependent connection strength is also evident in this dataset. This could add to the significance of the work and provide another angle to investigate circuit interactions. Again, this is only a suggestion.

We appreciate the reviewer’s suggestions. Indeed, we have previously classified the neurons described here into different categories based on their waveforms (Zhu et al., 2021, BioRxiv). We found that in addition to fast-spiking (FS) neurons, at least two-subclasses of regular-spiking (RS) neurons, namely regular-spiking medium (RS_M_) and regular-spiking long (RS_L_) neurons, exist within V1. This is generally consistent with evidence of the presence of more than two distinct classes of spike waveforms within mammalian cortex (e.g. Munoz, Tremblay and Rudy, 2014; Trainito, von Nicolai, Miller and Siegel, 2019) in addition to classes defined by burst propensity (Onorato et al., 2020). We found that the density of neurons from within these 3 putative waveform classes varied significantly across layers, and these laminar distributions were different among the 3 classes. Thus, although it would be interesting to examine whether fast-spiking and regular-spiking neurons exhibit differences in connection strength, we are cautious about overinterpreting the results from this analysis because (1) the 2 class FS vs RS categorization is probably an oversimplification of V1 population, and (2) the interactions between FS and RS neurons are likely confounded by their distinct laminar distributions and interactions with the local circuitry.

Moreover, although a number of previous studies have shown that fast-spiking (FS) neurons receive stronger and more dense input as compared to regular-spiking (RS) neurons, these studies were of thalamocortical connections and not cortico-cortical connections (Swadlow 2003; Callaway 2004; Cruikshank 2007; Bereshpolova, Hei, Alonso, and Swadlow 2020). As a result, in our data, the differences in the strength of input to RS and FS neurons may only be reflected indirectly in common-input pairs. It is upon future studies to more directly measure the strength of thalamic input to FS and RS neurons in monkey V1.

Nonetheless, as suggested by the reviewer, we examined differences in cell-type dependent connection strength between fast-spiking and regular-spiking neurons, separately for each cluster. We found that sharply synchronous pairs composed of two fast-spiking neurons (FS-FS) had significantly larger peak efficacies than sharply synchronous pairs composed of two regular-spiking neurons (RS-RS) (median peak efficacy: RS-RS = 0.030, FS-FS = 0.020; Wilcoxon rank sum test; p<10−5). In contrast, peak efficacy was not significantly different between FS-FS and RS-RS pairs in either the asynchronous or broadly synchronous classes (median peak efficacy: *Async*: RS-RS = 0.014, FS-FS = 0.016; *Bsync*: RS-RS = 0.022, FS-FS = 0.019; Wilcoxon rank sum test; *Async*, p=0.93,
*Bsync*, p=0.01). Although the difference in peak efficacy for sharply synchronous RS-RS and FS-FS pairs observed here may reflect a difference in the strength of thalamic input to FS and RS neurons, we did not add these results to the manuscript due to the aforementioned concerns.

4. It would be useful to show the CCGs in Figure 1e in a higher temporal resolution around the peak such that the lag of the peak is visible.

We agree and have edited the new Figure 1e to show the CCGs in a higher temporal resolution around the peak.

5. It seems odd to treat the forward and reverse correlation functions as two distinct types since the labeling is arbitrary. That is, swapping the label for the reference neuron and its partner would flip the correlation function from one category to the other. Since the choice of the reference neuron is arbitrary, it is not clear why there are two categories. Please provide justification.

We agree with the reviewer that the forward and reverse distinction is arbitrary, and in the revised paper, we have modified and clarified our treatment of forward and reverse CCGs. As now described, at the CCG clustering stage, there is no arbitrary choice regarding which neuron is the reference. As we clarify (Section of “*Classification of Cross-correlogram*” under Methods), the forward and reverse clusters emerge as a result of including both directions of the CCG. In other words, either neuron was chosen as the reference neuron. Thus, at the clustering stage, the number of ‘pairs’ was twice the number of significant CCGs. At this stage, we merely treat the forward and reverse distinction as expected from the clustering.

By contrast, in subsequent analyses where we compare the proportions of the different classes across layers or functional cell types, the choice of the reference neuron is done either randomly (Figures. 5a-c, 6b-c, 7c) or deliberately (Figures. 5d, 6d, 7b, Figure 5—figure supplement 1), e.g., in comparing the proportion of forward and reverse CCGs among simple-complex cell pairs, simple cells were chosen as the reference. These analyses, along with figures 5-7 have also been revised for emphasis and clarity. Crucially, all of the results remain unchanged.

6. Separate from this issue, the labeling looks erroneous to the best of our knowledge. That is, 'forward' correlation functions have more mass at negative lags. This is opposite to the conventional definition. Mass at negative lags means the partner neuron tends to fire before the reference neuron, which is not a forward connection (reference driving partner) but the opposite. Please clarify.

We thank the reviewer for noticing the mislabeling. We have corrected this in the revised manuscript.

7. Please provide the species used.

We have added the species details (*Macaca mulatta*) to the section of “*Experimental Model and Subject Details*” under Methods in the revised manuscript.

8. Page 10: It was not obvious that the regression coefficients could be directly compared since the covariates (pair distance and r_ori) have very different magnitudes. Please clarify.

We thank the reviewer for noting this. Indeed, the covariates of pair distance and r_ori_ have very different magnitudes. As a result, we have standardized (z-scored) these two independent variables, pair distance and r_ori_, prior to fitting the regression. Thus, the resulting regression coefficients can be interpreted as the change in the dependent variable associated with a one standard deviation (SD) change in the independent variable (see Kutner et al., *Applied Linear Statistical Models,* chpt. 7.5, pg. 272 for reference). Because we did not standardize the dependent variable, the units of the regression coefficients are the units of the dependent variable. For example, in the regression predicting peak lag, the coefficient of 0.94 for pair distance indicates that a one SD increase in the distance between two neurons is associated with a 0.94 ms increase in lag. We have amended the Results section “Dependence of synchrony and strength of functional interactions on tuning similarity” to clarify this.

9. Please provide an explanation of how the spike sorting was done. One concern about the sharp correlation peaks is that they are artifactual, resulting from issues in Kilosort (https://github.com/MouseLand/Kilosort/issues/29). In brief, Kilosort can match the same spike waveform to two different templates, one template capturing most of the waveform shape and the second capturing the residual. This issue is discussed on the Github Kilosort page. Are the authors sure that this does not contribute to/drive the sharp peak phenomenology? One way to provide reassurance would be to provide more information on the shape and width of the sharp peaks. If they are ~1 ms wide, it seems more likely to be artifactual than biological.

We thank the reviewer for pointing out the potential issue with spike sorting using Kilosort. Indeed, as the developer Marius Pachitariu acknowledged, this is an inherent problem using a template matching approach for spike sorting. Currently, there are no ideal solutions in Kilosort that fix this issue, so the users need to fix this themselves during the post-processing. We have added a section “*Data acquisition and spike sorting*” in the Methods to clarify this.

We adopted some common criteria in determining the occurrence of “double-counted spikes” (Siegle et al., 2021). For each pair of neurons with peak waveforms that are located within 50 μm (around 5 channels), we counted the number of spikes from one neuron that “overlap” with the other neuron, and then calculated the proportion of the “overlapping” spikes out of all the spikes in the given neuronal pair. Overlapping is defined as two spikes from the neuronal pair occurring within 5 samples (0.16 msec). We place no restriction on the amplitude relationship between the two neurons, although usually the double counted neuron from a residual exhibits much smaller amplitude.

**Author response table 1. sa2table1:** 

Percentage of overlapping spikes	>5%	>10%	>20%
N_pair_overlapped	26	16	7
N_pair_overlapped/N_pair_sharp synchronous	26/1408=1.85%	16/1408=1.14%	7/1408=0.5%

As Joshua Siegle mentioned *(*https://github.com/MouseLand/Kilosort/issues/29*)*, we should expect to see some perfectly overlapping spikes, just not very many. However, the exact boundary between true overlapping spikes and artificial overlapping spikes is not clear. In our case, we counted neuronal pairs within 50 μm that had at least 5%, 10%, and 20% overlapping spikes, and found that the number of pairs were 26, 16 and 7, respectively. Considering that we identified 1408 pairs (without considering directionality) that are sharply synchronous, those potentially artificial pairs contributed less than 2% of this class. Thus, although we didn’t remove these potentially contaminated pairs, we believe their contributions are negligible.

10. Please provide the proportion of simple and complex neurons in the data set.

The ratio of simple to complex neurons, respectively, was 1:2.4 in the dataset (236/802 neurons were simple, 566/802 neurons were complex). We have added this detail to the Results section (“Identifying functional interactions within single columns of visual cortex”) in the revised manuscript.

11. Discussion: The authors do a good job of discussing the limitations of correlation analysis for identifying synaptic connections. However, they refer to their R and F shapes (9% of cases) as potentially capturing such connections. Traditionally (e.g. Reid and Alonso), correlation function consistent with synaptic connections must pass a more stringent description than a simple asymmetry with respect to zero time lag. Namely, they need to have a sharp peak clearly offset from zero. This difference in stringency should be incorporated into the Discussion if the authors wish to propose that R and F types are related to connectivity.

We appreciate the reviewer’s suggestion and have revised the Discussion section to detail the variation in stringency of criteria used in past studies to identify monosynaptic connections (Section “*Functional interactions and synaptic connectivity*” under Discussions). The reviewer cites two such examples, namely the Liew et al. study (comment #2) and the Reid and Alonso studies (e.g., Reid and Alonso, 1995, 1999) (comment #11), which use different criteria. As we mention in the revised discussion, other, even more stringent, criteria have also been described, e.g., English et al., 2017; Senzai et al., 2019.

In the revised discussion, we also point out the obvious pitfalls of simply applying a more stringent criterion in order to identify monosynaptic connections. Although a key objective is to avoid false positives, the likelihood of false negatives is also a concern, particularly given that the variability in the parameters of such connections within a cortical circuit (e.g., synaptic latencies, synaptic strength) is not known. For example, consider the highly stringent criteria used by Buzsaki et al. (English et al., 2017; Senzai et al., 2019). Senzai et al., 2019 classified only ~0.5% of recorded pairs (525/102,332) as monosynaptic connections in mouse V1, which is considerably lower than the estimated rate of 5-12% “ground-truth” synaptic connections found using multi-patch slice recordings (e.g., Song et al., 2005; Cadwell et al., 2020). As the authors themselves explain in their discussion: “…detection of monosynaptic connections by the spike transmission probability method depends on the combination of the strength of the synapses and the number of spikes available in a given recording session, therefore it is biased toward neuronal pairs with stronger connections… This relationship may explain why only a small fraction of all possible E-E connections was detected” (Senzai., et al., 2019).

In addition, we note that the criteria used in the other example study cited by the reviewer (Liew et al., comment #2), is actually less stringent than ours in terms of the SD threshold. Using their criteria, which only requires the peak to be >3.5 SD above the shuffled background, our data yield 22,103 significant asynchronous connections. This is in stark contrast to the 6,124 significant asynchronous connections (asynchronous types) obtained with our 7 SD threshold, which is based on the threshold used by Siegle et al., 2021.

12. Line 635: The definition of the correlation function seems different from previous work. The correlation function appears normalized by a quantity that depends on the time lag of one neuron and not the other (the denominator). Please clarify the notation.

The difference is primarily notational, as we clarify here. The cross-correlation function used in previous work (e.g., Smith and Kohn 2005, 2008; Siegel et al. 2021) is,

CCG(τ) = 1M∑i=1M∑t=1T−τxi1(t) xi2(t+τ)(T−|τ|)λ1λ2 (Equation 1)

where M is the number of trials, T is the trial duration in milliseconds, xi1 is the spike train of neuron 1 in trial i, and λ1 is the firing rate of neuron 1. In this formulation, the cross-correlation is normalized by the product of the geometric mean of the firing rates of two neurons (λ1λ2) and a triangle function of lag (T−|τ|). This triangle function is a quantity that depends on the time lag of one neuron. Our definition of the normalization term is equivalent to that of previous work if the firing rates λ1 and λ2 are computed using only the bins used to compute the CCG at each time lag, i.e.,

λ1(τ)=∑i=1M∑t=1T−τxi1(t)(T−τ)×M and λ2(τ)=∑i=1M∑t=1T−τxi2(t+τ)(T−τ)×M.

This can be verified via substitution of λ1(τ) and λ2(τ) into Equation 1.

The difference between computing the firing rate normalization across all time bins or only time bins used to compute the CCG is typically negligible. However, the latter is more desirable when computing the CCG between neurons with non-stationary PSTHs that fluctuate at the beginning or end of a trial. To clarify the connection between our definition and that of previous work, we have modified the notation of the cross-correlation function in the revised manuscript (Section “*Cross-correlograms (CCGs*)” under Methods).

13. Figure 2B: There is an oscillation in the marginal histogram of peak delays. This is also evident in Figure 3C. Please clarify.

The peak delay is an integer between 0 and 10 ms for significant CCGs because CCGs were computed using spikes binned with 1 ms resolution. The oscillations in the marginal distribution were an artifact of applying a KDE function to this discrete distribution.

In the revised manuscript, we have fixed this in Figure. 2b, 3c, and 3e by illustrating raw histograms in the marginals instead of KDE functions of the histograms.